# Antigen-specific CD8+ T cell feedback activates NLRP3 inflammasome in antigen-presenting cells through perforin

Yikun Yao[1,2], Siyuan Chen[2], Mengtao Cao[2], Xing Fan[2], Tao Yang[2], Yin Huang[2], Xinyang Song[2], Yongqin Li[2], Lilin Ye[3], Nan Shen[2,4], Yufang Shi[2], Xiaoxia Li[5], Feng Wang[1] & Youcun Qian[1,2]

The connection between innate and adaptive immunity is best exemplified by antigen presentation. Although antigen-presenting cells (APCs) are required for antigen receptor-mediated T-cell activation, how T-cells feedback to APCs to sustain an antigen-specific immune response is not completely clear. Here we show that CD8+ T-cell (also called cytotoxic T lymphocytes, CTL) feedback activates the NLRP3 inflammasome in APCs in an antigen-dependent manner to promote IL-1β maturation. Perforin from antigen-specific CTLs is required for NLRP3 inflammasome activation in APCs. Furthermore, such activation of NLRP3 inflammasome contributes to the induction of antigen-specific antitumour immunity and pathogenesis of graft-versus-host diseases. Our study reveals a positive feedback loop between antigen-specific CTLs and APC to amplify adaptive immunity.

[1] Department of Nephrology, Shanghai Jiao Tong University Affiliated Sixth People's Hospital, Shanghai 200233, China. [2] Key Laboratory of Stem Cell Biology, CAS Center for Excellence in Molecular Cell Science, Institute of Health Sciences, Shanghai Institutes for Biological Sciences, Chinese Academy of Sciences & Shanghai Jiaotong University School of Medicine, Shanghai 200031, China. [3] Institute of Immunology, Third Military Medical University, Chongqing 400038, China. [4] Shanghai Institute of Rheumatology, Shanghai Renji Hospital, Shanghai Jiaotong University School of Medicine, Shanghai 200001, China. [5] Department of Immunology, Lerner Research Institute, Cleveland Clinic Foundation, Cleveland, Ohio 44195, USA. Correspondence and requests for materials should be addressed to F.W. (email: Zyzwq1030@163.com) or to Y.Q. (email: ycqian@sibs.ac.cn).

NACHT, LRR and PYD domains-containing protein 3 (NLRP3) is the most studied member of the Nod-like receptor (NLR) family. NLRP3 is activated primarily in innate immune cells such as dendritic cells and macrophages, and by a variety of stimuli, including pathogens and danger signals such as monosodium urate (MSU) and ATP[1–5]. Upon stimulation, NLRP3 recruits the adaptor Apoptosis-associated Speck-like protein containing a CARD (ASC) through PYD–PYD domain association, and ASC further recruits caspase-1 through CARD–CARD domain interaction, forming the signalling complex known as the inflammasome. Activated caspase-1 then cleaves pro-IL-1β to form mature IL-1β with pro-inflammatory functions[3,4]. In addition to caspase-1, bacterial infections also activate caspase-11 for the 'non-canonical' NLRP3 inflammasome pathway[6,7]. Dysregulation of NLRP3 inflammasome activation is associated with a variety of inflammatory disorders, such as cryopyrin-associated periodic syndromes and diabetes[8–11]. However, the functions of NLRP3 inflammasome in the pathogenesis of tumours and graft-versus-host disease (GVHD) are less defined[12–14] and it is unclear whether the NLRP3 inflammasome has a function in antigen-specific antitumour immunity.

Antigen-presenting cells (APCs) bridge innate and adaptive immunity. Antigens are processed and presented in APCs through MHC class II or MHC class I to activate naïve CD4[+] or CD8[+] T cells, respectively[15]. β2 microglobulin (β2M) is a subunit of MHC class I and has been shown to be required for antigen-specific CD8[+] T cells (also called cytotoxic T lymphocytes, CTLs) differentiation, activation and proliferation[16]. Antigen-activated CTLs have critical functions in host defense against tumours and pathogens, as well as in the pathogenesis of GVHD[17]. The cytolytic killing of target cells by CTLs requires perforin-mediated release of granzymes, mainly granzyme B, from cytotoxic granules[18–20]. Fas-FasL signalling also contributes to CTL-mediated effects[21].

Although innate immunity instructs adaptive immunity for antigen-specific immune responses, adaptive immunity has also been shown to suppress innate immunity to modulate abnormal inflammatory responses during viral infection in an antigen-independent manner[22]. T regulatory (Treg) cells are well-defined suppressors of both adaptive and innate effector cells and function via the secretion of suppressive cytokines or by cell–cell contact[23]. One study reported that anti-CD3-activated T cells dampen innate immune responses through suppressing the NLRP3 inflammasome in macrophages in an antigen-independent manner[24]. However, it is not completely clear how innate immunity-driven adaptive immunity feedback promotes innate immunity to amplify antigen-specific immune responses. Here, we show that CTLs activate the NLRP3 inflammasome in APCs which amplifies antigen-specific CTL-mediated effector functions.

## Results

**Inflammasome assembly induced by antigen-specific CTLs.** ASC is a key adaptor of several inflammasomes such as NLRP3 and AIM2, and its activation is reflected by ASC speck assembly or oligomerization[4]. We utilized ASC speck assembly as a readout to search for potential new ASC inflammasome activators and found that OT1 CTLs induced ASC speck assembly in bone marrow-derived dendritic cells (BMDCs) pulsed with OVA peptide during co-culture, similar to MSU treatment (Fig. 1a,b). We also found that OT1 CTLs induced ASC oligomerization (Fig. 1c). Consistent with the ASC activation, we observed that OT1 CTLs activated Caspase-1 and consequently induced IL-1β maturation and secretion (Fig. 1c,d). However, protein levels of IL-6 and pro-IL-β were not induced (Supplementary Fig. 1a;

Fig. 1c), indicating that inflammasome activation is responsible for IL-1β secretion. Kinetic experiments showed that OT1 CTLs induced IL-1β secretion as early as 1 h after the incubation (Fig. 1e). OT1 CTLs also induced IL-1β secretion in OVA-pulsed bone marrow-derived macrophages (BMDM) or peritoneal macrophages (PMs) (Fig. 1f; Supplementary Fig. 1b,c). CTLs are the primary killer cells in the mixed-lymphocyte reaction (MLR) assay in an antigen-specific manner[25]. Similar to OT1 CTLs, CTLs from the MLR induced ASC speck assembly in BMDCs (Fig. 1g,h). The CTLs also induced ASC oligomerization, caspase-1 activation and consequently IL-1β maturation and secretion (Fig. 1i,j; Supplementary Fig. 1d). We then compared the ability of CTLs and the other remaining cells in the MLR to induce IL-1β production in BMDCs and found that CTLs were the predominant cells for IL-1β secretion and target cell killing (Fig. 1k; Supplementary Fig. 1e). Alloantigen-specific CTLs can be determined by loss of CFSE staining in MLRs[26]. Therefore, we isolated CFSE[high] CTLs from MLR as non-responding cells and CFSE[low] CTLs from MLR as alloantigen-specific CTLs, and then co-cultured them with allogeneic DC cells and we found that alloantigen-specific CTLs (CFSE[low] CTLs) induced IL-1β secretion (Fig. 1l; Supplementary Fig. 1f). Together, our data suggest that antigen-specific CTLs induce the ASC-mediated inflammasome assembly for IL-1β maturation and secretion in APCs.

**NLRP3 is essential for CTL-induced IL-1β secretion.** As antigen-specific CTLs induced ASC oligomerization for IL-1β secretion in APCs (Fig. 1), we wanted to determine which inflammasome was important for ASC activation and IL-1β maturation. We found that IL-1β secretion induced by antigen-specific OT1 CTLs was nearly blocked in Nlrp3-deficient BMDCs but was not affected in Aim2-deficient BMDCs (Fig. 2a,b; Supplementary Fig. 2a), suggesting that OT1 CTLs activate NLRP3 inflammasome for IL-1β secretion. Indeed, we found that OT1 CTLs induced co-localization of NLRP3 and ASC. Conversely, OT1 CTL-induced ASC speck assembly was not observed in Nlrp3-deficient BMDCs (Fig. 2c). Similarly, antigen-specific OT1 CTL-induced ASC oligomerization and caspase-1 activation were lost in Nlrp3-deficient BMDCs (Fig. 2d). Consistent with the requirement of NLRP3 for caspase-1 activation, OT1 CTL-induced IL-1β secretion was almost blocked in Caspase1-deficient BMDCs (Fig. 2e; Supplementary Fig. 2b). Furthermore, the caspase-1-specific inhibitor impaired OT1 CTL-induced IL-1β secretion (Fig. 2f), indicating that caspase-1 activation is required for CTL-mediated IL-1β maturation. We further found that caspase-11 was not required for OT1 CTL-induced IL-1β secretion by utilizing 129X1/SvJ mice with a caspase-11 mutation and deficiency[6] (Fig. 2g; Supplementary Fig. 2c). In addition to BMDCs, NLRP3 was also required for OT1 CTL-induced ASC oligomerization, caspase-1 activation and IL-1β secretion in BMDMs pulsed with OVA (Fig. 2h,i). To determine whether NLRP3 is involved in antigen-specific CTL-induced IL-1β secretion in APCs in vivo, Nlrp3-deficient and wild-type control mice were challenged with OVA (intraperitoneal (i.p.)) to prime antigen presentation followed by an injection of OT1 CTLs. We found that NLRP3 deficiency indeed significantly reduced IL-1β production as well as its mediated neutrophil influx (Fig. 2j).

**Antigen-dependent NLRP3 activation by CTLs.** We found that OT1 CTLs induced NLRP3 inflammasome activation and IL-1β secretion in APCs pulsed with OVA but not in APCs without an OVA challenge (Figs 1 and 2). To confirm that the CTL-induced NLRP3 inflammasome activation requires antigen

presentation, we utilized *β2M*-deficient BMDCs with a defect in antigen presentation. OT1 CTL-induced ASC speck assembly was not observed in *β2M*-deficient BMDCs pulsed with OVA (Fig. 3a). Furthermore, antigen-specific OT1 CTL-induced ASC oligomerization and caspase-1 activation were blocked in *β2M*-

deficient BMDCs (Fig. 3b). Consistently, β2M deficiency blocked antigen-specific OT1 CTL-induced IL-1β secretion (Fig. 3c; Supplementary Fig. 2d). Similarly, OT1 CTL-induced IL-1β secretion was almost blocked in OVA-pulsed *H2-KbDb*$^{-/-}$ BMDCs, as H2-Kb specifically recognizes OVA peptide (Fig. 3d).

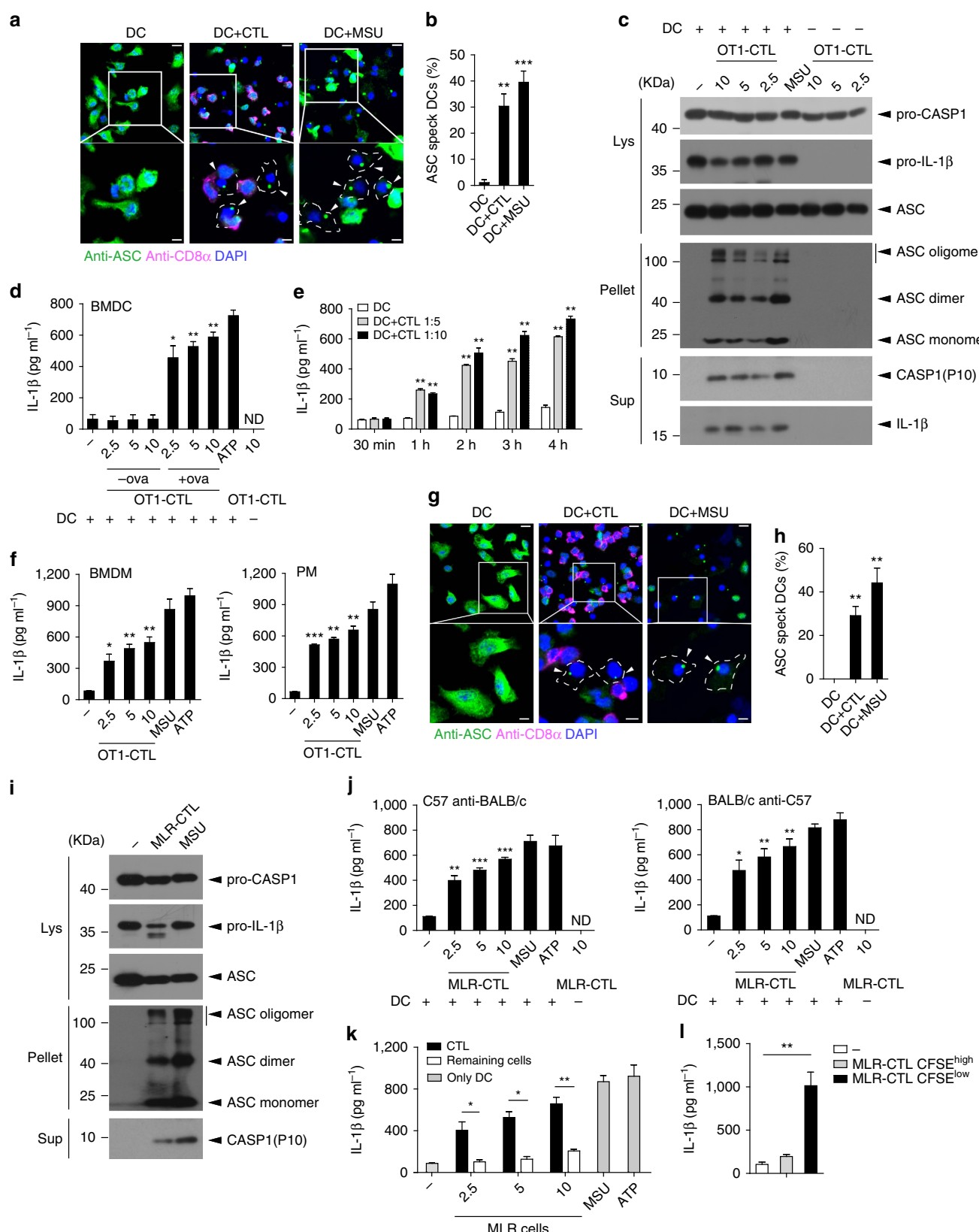

Similar to the OT1 system, MLR-CTL-induced ASC speck formation was not observed in β2M-deficient BMDCs (Fig. 3e). β2M deficiency also blocked MLR-CTL-induced ASC oligomerization, caspase-1 activation and consequently IL-1β secretion (Fig. 3f,g; Supplementary Fig. 2e). We further utilized concanavalin A (ConA), PMA plus ionomycin or anti-CD3 to activate purified CTLs in an antigen-independent manner and found that these activated CTLs did not induce IL-1β secretion, although they induced IFN-γ secretion (Fig. 3h,i; Supplementary Fig. 2f). To confirm these results *in vivo*, we immunized OT1 mice with OVA peptide which primed APCs and activated OT1 cells *in vivo*. We found that the first OVA challenge induced IL-1β secretion in the spleen, and the second OVA challenge had a stronger effect (Fig. 3j). We further confirmed that OT1 CTL-induced IL-1β secretion *in vivo* requires antigen presentation as IL-1β secretion by OT1 CTLs was significantly reduced in *β2M-* or *H2-KbDb*-deficient mice (Fig. 3k). These results indicate that MHC I-mediated antigen presentation by APCs activates CTLs, and the activated antigen-specific CTLs feedback activate NLRP3 inflammasome for IL-1β secretion in APCs.

**Perforin from CTLs modulates IL-1β secretion by APC.** The danger signal ATP that activates NLRP3 inflammasome is secreted from dead cells[14]. To determine whether any potential soluble factors secreted by cells are important for antigen-specific CTL-induced IL-1β secretion, we collected cell culture medium (supernatant) of the mixed OT1 CTLs and BMDCs and used the supernatant to treat BMDCs. We found that the supernatant did not further induce IL-1β secretion in BMDCs (Fig. 4a), indicating that cell–cell contact is likely required for OT1 CTL-induced IL-1β secretion. We further utilized transwell experiments to confirm that cell–cell contact is required for OT1 CTLs to induce IL-1β secretion from OVA-pulsed BMDCs (Fig. 4b) or BMDMs (Supplementary Fig. 3a). Fas-FasL signalling is reported to induce non-canonical IL-1β maturation[27]. We found that OT1 or MLR CTL-induced IL-1β secretion was not significantly reduced in *Fas*-deficient (*Fas^lpr*) BMDCs (Fig. 4c; Supplementary Fig. 3b), indicating that Fas signalling is not important for CTL-mediated IL-1β maturation. To determine whether cell apoptosis contributes to the observed IL-1β secretion, we compared the kinetics of cell apoptosis with that of IL-1β secretion and found that both target cell apoptosis (Supplementary Fig. 3c) and IL-1β secretion (Fig. 1e) occurred at 1 h while cell death occurred at 2 h (Supplementary Fig. 3c). We then knocked down Casp8 and RIP3, the key signalling molecules in cell apoptosis and necroptosis, respectively, and found that the two molecules were not required for OT1 CTL-induced IL-1β secretion in BMDMs (Supplementary Fig. 3d,e). Similarly, IL-1β secretion was

not affected by a combination of apoptosis and necroptosis inhibitors (Supplementary Fig. 3f,g) or by RIP3 deficiency plus an apoptosis inhibitor (Supplementary Fig. 3h,i). These data suggest that target cell apoptosis and necroptosis are not required for antigen-specific CTL-induced IL-1β secretion in APCs.

The cytolytic killing of target cells by CTLs requires perforin-mediated release of granzymes from cytotoxic granules[28]. We found the expression of perforin and granzymes was induced in CTLs after antigen-mediated activation, and the expression of granzyme B was much higher than that of other granzymes (Supplementary Fig. 4a). We next checked perforin and granzyme B levels in naive, effector memory, and central memory CTLs. We found effector memory CTLs comprise the majority of the population of CTLs (Fig. 4d) and had the highest expression of perforin and granzyme B in the MLR (Fig. 4e). Similarly, the effector memory population of OT1 CTLs had the highest expression of perforin and granzyme B, and their expression levels were highly induced in OT1 CTLs by OVA peptide stimulation (Supplementary Fig. 4b). Notably, we observed that the effector memory CTLs had the strongest ability to induce IL-1β secretion in APCs of the populations of the MLR and OT1 CTLs (Fig. 4f; Supplementary Fig. 4c), and the expression levels of perforin and granzyme B in the different MLR-CTL or OT1-CTL populations correlated with their induced IL-1β secretion levels in APCs (Fig. 4e,f; Supplementary Fig. 4b,c). We then utilized a perforin inhibitor as reported previously[29,30] and found that the inhibitor suppressed IL-1β secretion (Fig. 4g; Supplementary Fig. 4d), suggesting that perforin is likely important for CTL-induced IL-1β production. To confirm this, we utilized *Perforin*-deficient mice and found that perforin deficiency in CTLs almost blocked IL-1β secretion and cell death in BMDCs (Fig. 4h; Supplementary Fig. 4e). We then determined the potential contribution of granzyme B and found that deficiency of granzyme B in CTLs also reduced IL-1β secretion, while the granzyme B-mediated effect was much weaker than that of perforin (Fig. 4h). Furthermore, double deficiency of granzyme A and B in CTLs had an effect on IL-1β secretion similar to single granzyme B deficiency (Fig. 4h), indicating that granzyme A is not important for MLR CTL-induced IL-1β production. Consistent with the very weak expression of perforin and granzymes in BMDCs (Supplementary Fig. 4a), deficiency of perforin, granzyme B or granzymes A and B in BMDCs did not affect the specific CTL-mediated induction of IL-1β and cell death (Supplementary Fig. 4f). However, it is reported that *Pasteurella multocida* toxin induces granzyme A expression in macrophage and the exocytosed granzyme A enters target cells and mediates IL-1β maturation independently of caspase-1 and without inducing cytotoxicity[31]. The differential use of granzyme A in IL-1β maturation is probably due to the different experimental systems.

**Figure 1 | Inflammasome assembly in APCs induced by antigen-specific CTLs.** (**a,b**) Confocal microscopy analysis (**a**) and quantification (**b**) of ASC specks assembled of LPS-primed and ova-pulsed BMDCs incubated with activated OT1-CTLs (DC + CTL) or with MSU (DC + MSU). Scale bar, 10 μm of the top panel, 5 μm of the bottom panel. (**c**) Western blot of cell lysates (Lys), DSS cross-linked pellets (Pellet) or supernatants (Sup) from LPS-primed and ova-pulsed BMDCs co-cultured with activated OT1-CTLs at the ratios indicated for 4 h. (**d**) The levels of IL-1β from LPS-primed BMDCs that were pulsed with or without ova peptide for 1 h and then co-cultured with the activated OT1-CTLs at the ratios indicated. LPS + ATP (ATP) was used as a positive control. (**e**) IL-1β release from LPS-primed and ova-pulsed BMDCs co-incubated with OT1-CTLs for the indicated time. (**f**) The levels of IL-1β from LPS-primed and ova-pulsed BMDMs or peritoneal macrophages (PM) co-incubated with OT1-CTLs for 4 h. (**g,h**) Confocal microscopy analysis (**g**) and quantification (**h**) of ASC specks assembled of LPS-primed BMDCs incubated with MLR-CTLs (DC + CTL) or with MSU (DC + MSU). Scale bar, 10 μm of the top panel, 5 μm of the bottom panel. (**i**) Western blot of cell lysates (Lys), DSS cross-linked pellets (Pellet) or supernatants (Sup) from LPS-primed BMDCs co-cultured with MLR-CTLs. (**j**) IL-1β release determined by ELISA from LPS-primed BMDCs co-cultured with MLR-CTLs (C57 anti-BALB/c or BALB/c anti-C57) with the indicated ratios for 4 h. MSU and ATP were used as positive controls. (**k**) The levels of IL-1β from LPS-primed BMDCs co-cultured with purified MLR CTLs or all the other remaining cells for 4 h. (**l**) The level of IL-1β from LPS-primed and ova-pulsed BMDCs co-cultured with MLR-CTLs (C57 anti-BALB/c) that were sorted as indicated in Supplementary Fig. 1f. Data are representative of two (**c,e,i**) or three (**a,b,d,f–h,j–l**) independent experiments. Error bars, s.e.m. *P < 0.05, **P < 0.01, ***P < 0.001 by two-tailed Student's t-test.

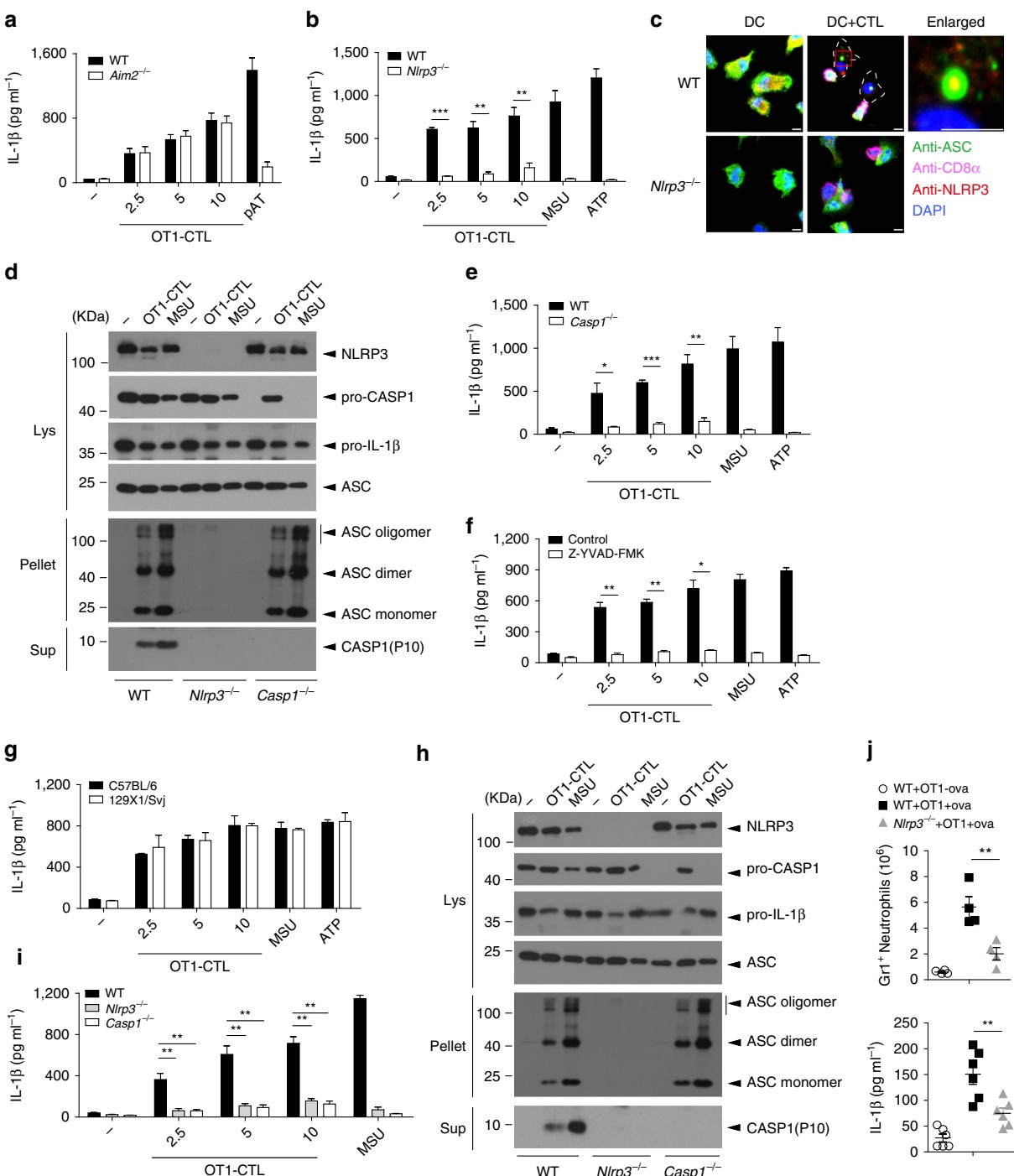

**Figure 2 | The essential role of NLRP3 for CTL-induced IL-1β secretion.** (**a**) IL-1β release from LPS-primed and ova-pulsed wild-type or *Aim2*[−/−] BMDCs co-incubated with OT1-CTLs at the ratios indicated for 4 h. pAT (poly(dA:dT)) was used as a positive control. (**b**) The levels of IL-1β from LPS-primed and ova-pulsed wild-type or *Nlrp3*[−/−] BMDCs co-incubated with OT1-CTLs at the ratios indicated. MSU and ATP were used as positive controls. (**c**) Confocal microscopy analysis of ASC specks assembled of LPS-primed and ova-pulsed wild-type (WT) or *Nlrp3*[−/−] BMDCs incubated with activated OT1-CTLs (DC + CTL). Far right, magnification of the area in red colour outlined at left showing the typical NLRP3 inflammasome structure. Scale bar, 5 μm. (**d**) Western blot analysis of cell lysates (Lys), DSS cross-linked pellets (Pellet) or supernatants (Sup) of wild-type, *Nlrp3*[−/−] or *Casp1*[−/−] BMDCs that were LPS-primed and ova-pulsed and then co-cultured with OT1-CTLs. (**e**) The levels of IL-1β from LPS-primed and ova-pulsed wild-type or *Casp1*[−/−] BMDCs co-incubated with OT1-CTLs at the ratios indicated. (**f**) IL-1β release from LPS-primed and ova-pulsed BMDCs co-incubated with OT1-CTLs in the presence or absence of the Caspase-1 inhibitor z-YVAD-fmk for 4 h. (**g**) The levels of IL-1β from wild-type C57 or 129X1/SvJ BMDCs that were LPS-primed and ova-pulsed and then co-cultured with OT1-CTLs. (**h,i**) Western blot analysis of cell lysates (Lys), DSS cross-linked pellets (Pellet) or supernatants (Sup) (**h**) and IL-1β release (**i**) of wild-type, *Nlrp3*[−/−] or *Casp1*[−/−] BMDMs that were LPS-primed and ova-pulsed and then co-cultured with OT1-CTLs. (**j**) Neutrophil influx or IL-1β level of peritoneal lavage from wild-type or *Nlrp3*[−/−] mice that were challenged i.p. with or without ova (10 μg) for 1.5 h and then with OT1-CTLs for 6 h. *n* = 4 Neutrophil influx, *n* = 6 IL-1β level. Data are representative of two (**d,f,g,h**) or three (**a–c,e,i,j**) independent experiments. Error bars, s.e.m. *$P < 0.05$, **$P < 0.01$, ***$P < 0.001$ by two-tailed Student's *t*-test.

In contrast to our study showing that antigen-specific CTLs induced NLRP3 inflammasome in APCs (Figs 1–3), one study reported that *in vitro* anti-CD3-activated CTLs suppress NLRP3 inflammasome in macrophages in an antigen-independent manner[24]. We also confirmed that anti-CD3-activated CTLs inhibited the MSU-induced NLRP3 inflammasome in BMDMs (Fig. 4i). Notably, we observed that perforin deficiency in CTLs did not affect their suppressive effect on MSU-induced IL-1β

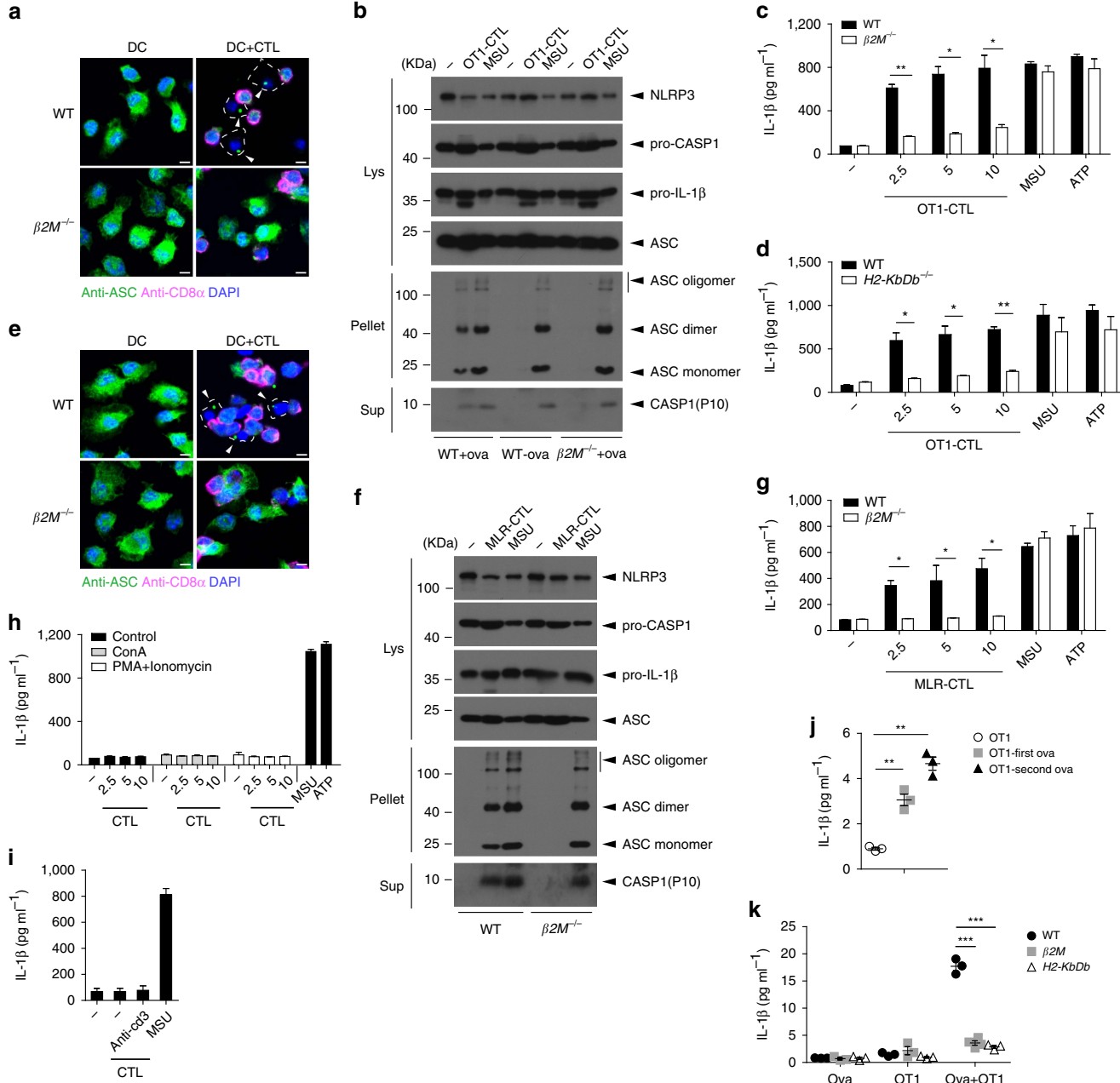

**Figure 3 | Antigen presentation for CTL-induced NLRP3 activation.** (**a**) Confocal microscopy analysis of ASC specks assembled of LPS-primed and ova-pulsed wild-type (WT) or $\beta 2M^{-/-}$ BMDCs incubated with activated OT1-CTLs (DC + CTL). Scale bar, 5 μm. (**b**) Western blot of cell lysates (Lys), DSS cross-linked pellets (Pellet) or supernatants (Sup) from LPS-primed and with or without ova-pulsed wild-type BMDCs or LPS-primed and ova-pulsed $\beta 2M^{-/-}$ BMDCs co-cultured with activated OT1-CTLs. (**c**) IL-1β release from LPS-primed and ova-pulsed BMDCs from wild-type or $\beta 2M^{-/-}$ mice that were co-cultured with OT1-CTLs. (**d**) IL-1β release from LPS-primed and ova-pulsed BMDCs from wild-type or $H2\text{-}KbDb^{-/-}$ mice that were co-cultured with OT1-CTLs. (**e**) Confocal microscopy analysis of ASC specks assembled of LPS-primed wild-type (WT) or $\beta 2M^{-/-}$ BMDCs incubated with MLR-CTLs (DC + CTL) (BALB/c anti-C57). Scale bar, 5 μm. (**f**) Western blot of cell lysates (Lys), DSS cross-linked pellets (Pellet) or supernatants (Sup) from LPS-primed wild-type BMDCs or $\beta 2M^{-/-}$ BMDCs co-cultured with MLR-CTLs. (**g**) IL-1β release from LPS-primed BMDCs from wild-type or $\beta 2M^{-/-}$ mice that were co-cultured with MLR-CTLs. (**h,i**) IL-1β levels from LPS-primed BMDCs co-cultured with ConA, PMA plus ionomycin (**h**) or anti-CD3 (**i**) activated CTLs. (**j**) OT1 mice were challenged i.v. with ova peptide (2 μg) once (OT1-first ova) or rechallenged i.v. on day 7 (OT1-second ova). IL-1β release from spleen homogenates from challenged and rechallenged mice was determined by ELISA after 6 h i.v. with ova peptide. $n = 3$ per group. (**k**) IL-1β release from spleen homogenates from wild-type, $\beta 2M^{-/-}$ or $H2\text{-}KbDb^{-/-}$ mice that were challenged i.v. with ova peptide for 1.5 h and then injected i.v. with activated OT1-CTLs for 6 h. $n = 3$ per group. Data are representative of two (**b–d,f**) or three (**a,e,g–k**) independent experiments. Error bars, s.e.m. *$P < 0.05$, **$P < 0.01$, ***$P < 0.001$ by two-tailed Student's $t$-test.

secretion in an antigen-independent manner (Fig. 4i), suggesting that perforin is specifically required for CTL-mediated IL-1β secretion in APCs in an antigen-dependent manner.

There are two models for perforin functions. The first model for perforin and granzymes mediated killing is that granzymes enter target cells via plasma membrane pores formed by perforin[18,32]. The second suggests that perforin forms transient pores in the plasma membrane to induce the endocytosis of perforin and granzymes into the target cells and then forms pores in endosomes to trigger cytosolic release[18,33,34]. We next determined whether perforin can induce IL-1β secretion in APCs. Perforin needs to be dissolved in calcium containing

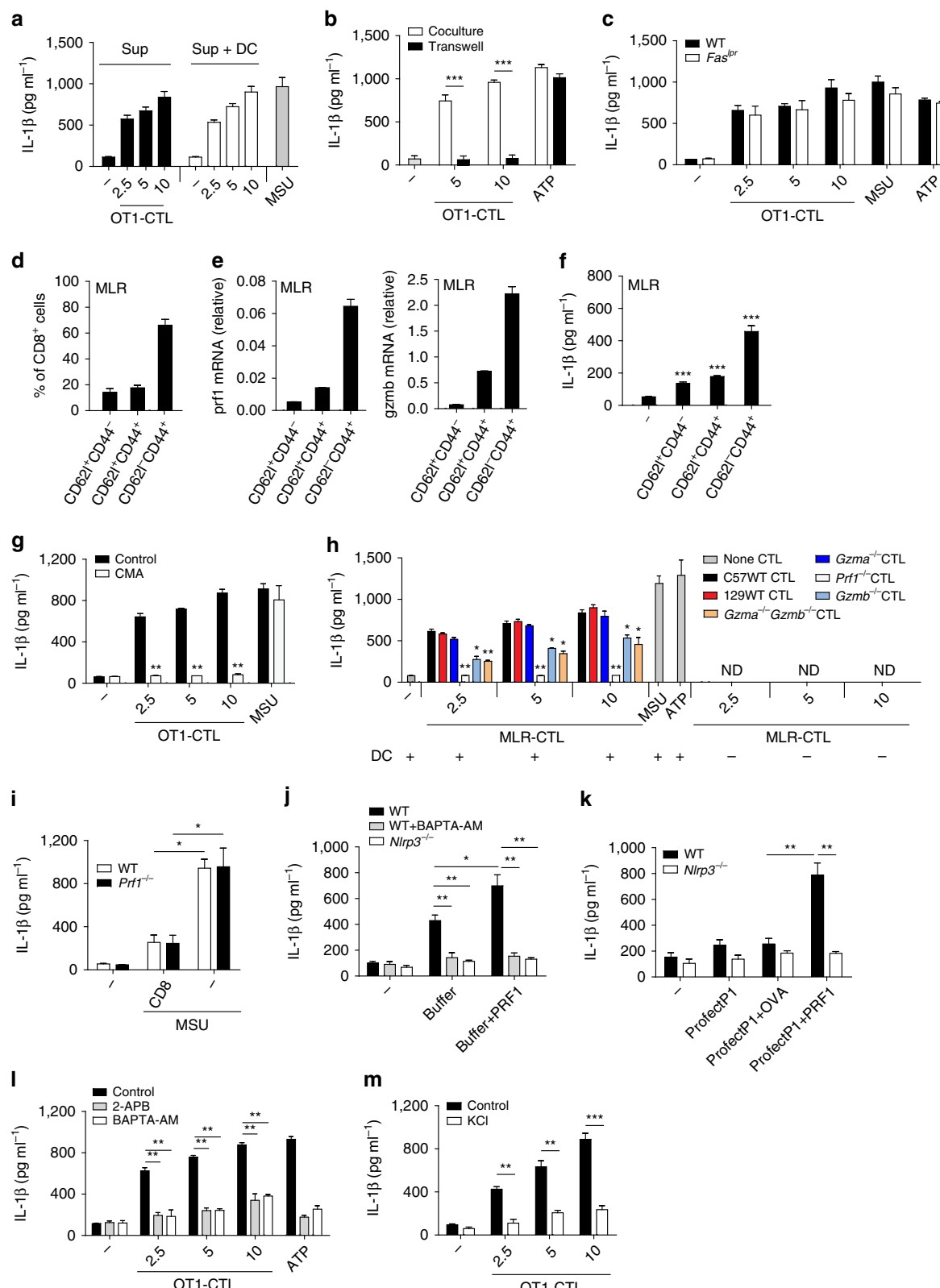

perforin buffer when added to medium to treat target cells, and calcium is important for the function of perforin[34–36]. We found that IL-1β secretion was already highly induced in the supernatant of BMDCs that were treated with only perforin buffer (Fig. 4j), likely due to the high extracellular calcium concentration in the buffer because calcium is reported to be essential for NLRP3 activation[37] and the calcium inhibitor or NLRP3 deficiency blocked the buffer-induced IL-1β secretion (Fig. 4j). Under this condition, perforin still statistically increased IL-1β secretion although the increase was not dramatic (Fig. 4j), and the increase of IL-1β secretion by perforin was blocked by the calcium inhibitor or NLRP3 deficiency (Fig. 4j). We also utilized the profectP1 protein delivery system and found that perforin delivered to the BMDCs markedly induced IL-1β secretion and NLRP3 was required for perforin-induced IL-1β secretion under this delivery system (Fig. 4k). We then explored how perforin affects NLRP3 inflammasome activation for IL-1β maturation and secretion. As delivery of perforin to target cells by antigen-specific CTLs has been reported to induce calcium influx into the cytosol[38], and calcium has been shown to be required for NLRP3 inflammasome activation[37], we determined the effect of calcium on antigen-specific CTL-induced IL-1β production in APCs. The endoplasmic reticulum (ER) is the major source of intracellular calcium influx under stress conditions[37]. We checked expression levels of ER $Ca^{2+}$ channels, including IP3Rs and RyRs, in BMDCs and found that IP3Rs were highly expressed (Supplementary Fig. 4g). We further observed that either inhibition of calcium influx from ER by the inhibitor 2-APB for IP3Rs or depletion of intracellular calcium by BAPTA-AM blocked CTL-induced IL-1β secretion in BMDCs in an antigen-dependent manner (Fig. 4l; Supplementary Fig. 4h). In addition to calcium influx, potassium efflux is also required for NLRP3 inflammasome activation by multiple triggers[39]. We found that extracellular potassium chloride also suppressed OT1 CTL-induced IL-1β secretion (Fig. 4m; Supplementary Fig. 4i). As perforin delivered to target cells forms membrane pores[18,34], these data suggest that the pore-forming ability of perforin likely induces potassium efflux and calcium influx to activate NLRP3 inflammasome.

**CTL-activated NLRP3 inflammasome in antitumour immunity.** CTLs play an important role in killing tumour cells through tumour antigen-specific immunity[40]. To understand whether CTLs induce IL-1β production during antigen-specific antitumour immunity, we established an *in vivo* model in which mice were injected subcutaneously (s.c.) with the tumour cell line EL4 (a T-cell-derived tumour cell line) or OVA containing EL4 (EG7) for tumour formation, and then injected intravenously (i.v.) with activated OT1 CTLs to kill EG7 target cells (as depicted in Supplementary Fig. 5a). We found that OT1 CTLs quickly induced IL-1β secretion and IFN-γ production in the EG7 tumours but not in EL4 tumours (Fig. 5a; Supplementary Fig. 5b). Furthermore, OT1 CTL-induced IL-1β and IFN-γ production was blocked in *β2M*-deficient mice (Fig. 5b; Supplementary Fig. 5c), indicating that antigen presentation is required for IL-1β secretion *in vivo*. To determine whether the NLRP3 inflammasome activated by tumour antigen-specific CTLs contributes to their mediated antitumour immunity, we utilized a similar strategy (as depicted in Supplementary Fig. 5d) to analyse the ability of OT1-mediated antigen-specific killing of target tumour cells. As expected, OT1 rejected EG7 tumours but not EL4 tumours (Supplementary Fig. 5e,f), and IL-1β secretion was increased during OT1-mediated antigen-specific rejection of EG7 tumours (Supplementary Fig. 5g). Importantly, deficiency of NLRP3 or Caspase-1 significantly blocked the OT1 CTL-mediated rejection of EG7 tumours (Fig. 5c,d). Consistently, we found that IL-1β secretion was blocked in *Nlrp3-*, *Caspase-1-* and *β2M*-deficient mice (Fig. 5e), indicating that antigen-specific activation of NLRP3 inflammasome is required for IL-1β maturation in the CTL-mediated tumour rejection model. IL-1β has been reported to affect CTLs priming and activation[41] and tumour antigen-activated CTLs are crucial for tumour clearance. We observed that IFN-γ producing CTLs were not detected by deficiency of β2M and were partially blocked by deficiency of NLRP3 or Caspase-1 in tumour model (Fig. 5f). DC cells from WT, *Nlrp3-*, *Caspase-1-*deficient mice had a comparable MHC I level while β2M deficiency reduced MHC I expression (Supplementary Fig. 5h), indicating that the reduction of IFN-γ producing CTLs of NLRP3 or Caspase-1 deficiency is likely due to IL-1β decrease but not due to the differences in antigen load while the reduction of IFN-γ producing CTLs of β2M deficiency is due to differences in antigen load. We previously showed that perforin was essential for CTL-induced IL-1β secretion *in vitro* (Fig. 4h). Therefore, *Perforin*-deficient OT1 CTLs were injected to estimate the contribution of perforin in the tumour model. We found deficiency of perforin significantly blocked the OT1 CTL-mediated rejection of EG7 tumours (Fig. 5g,h) and also reduced IL-1β secretion (Fig. 5i). We have showed above that effector memory CTLs were the majority population to induce IL-1β secretion in APCs *in vitro* (Supplementary Fig. 4b,c). Effector memory CTLs have been reported to migrate to peripheral tissues and perform immediate effector functions, while naïve and central memory CTLs usually home to secondary lymphoid organs and have little or no effector functions[42]. Therefore, we analysed CTL populations in the tumours and found that effector memory CTLs were the majority of the population and had the highest expression of perforin and granzyme B (Supplementary Fig. 5i,j), suggesting that effector memory CTLs are critical for IL-1β secretion in the tumour

**Figure 4 | Perforin in CTLs required for IL-1β secretion in APCs.** (**a**) IL-1β release in the supernatants (Sup) of LPS-primed and ova-pulsed BMDCs co-cultured with OT1-CTLs for 4 h or in the supernatants (Sup + DC) from LPS-primed DC cells stimulated with the previous Sup for 4 h. (**b**) IL-1β release from LPS-primed and ova-pulsed BMDCs, co-cultured with OT1-CTLs or with transwell-seperated OT1-CTLs for 4 h. LPS + ATP (ATP) were added as a positive control. (**c**) IL-1β release from LPS-primed wild-type or *Fas^lpr* BMDCs co-cultured with OT1-CTLs. (**d**) The percentages of different CTL populations as indicated in MLR reaction at day 6. (**e**) The mRNA levels of perforin and granzyme B of CTL populations as in **d**. (**f**) IL-1β secretion from BMDCs co-cultured with the different CTL populations from MLR as in **e**. (**g**) IL-1β release from LPS-primed and ova-pulsed BMDCs co-incubated with OT1-CTLs for 4 h in the presence of perforin inhibitor CMA (100 ng ml$^{-1}$). (**h**) IL-1β release from LPS-primed BMDCs co-cultured with MLR-CTLs (C57 or 129 anti BALB/c) from the indicated mice. (**i**) IL-1β release from BMDMs that were co-cultured overnight with anti-CD3 plus anti-CD28 treated CTLs from wild-type and *Prf1^−/−* mice in the presence of anti-CD3 and then were primed with LPS and stimulated with MSU. (**j**) IL-1β release from LPS-primed wild-type or *Nlrp3^−/−* BMDCs treated with the perforin buffer or perforin buffer plus perforin in the presence or absence of calcium inhibitor BAPTA-AM (50 μM). (**k**) IL-1β release from LPS-primed wide-type or *Nlrp3^−/−* BMDCs transfected with the indicated proteins by the profectP1 protein delivery system. (**l**) IL-1β release from LPS-primed ova-pulsed BMDCs co-cultured with OT1 CTLs in the presence of 2-APB (50 μM) or BAPTA-AM (25 μM). (**m**) IL-1β release from LPS-primed ova-pulsed BMDCs co-cultured with OT1 CTLs in the presence or absence of KCl (50 mM). Data are representative of three (**a–m**) independent experiments. Error bars, s.e.m. *P < 0.05, **P < 0.01, ***P < 0.001 by two-tailed Student's t-test.

model. We then determined whether antigen-specific CTL-induced IL-1β secretion contributes to antitumour immunity by utilizing *Il-1r1*-deficient mice. Indeed, the receptor deficiency resulted in significant blockade of OT1 CTL-mediated antigen-specific antitumour immunity (Fig. 5j,k). Consistently, antigen-specific CTL activation was reduced in the receptor-

deficient mice (Supplementary Fig. 5k). To identify the cellular source of IL-1β secretion, we first examined the expression levels of NLRP3 inflammasome components in APCs (BMDCs and BMDMs) and EG7 tumour cells and found that NLRP3 and pro-IL-1β were not detected in EG7 cells, even after LPS stimulation (Supplementary Fig. 5l). Consistently, OT1

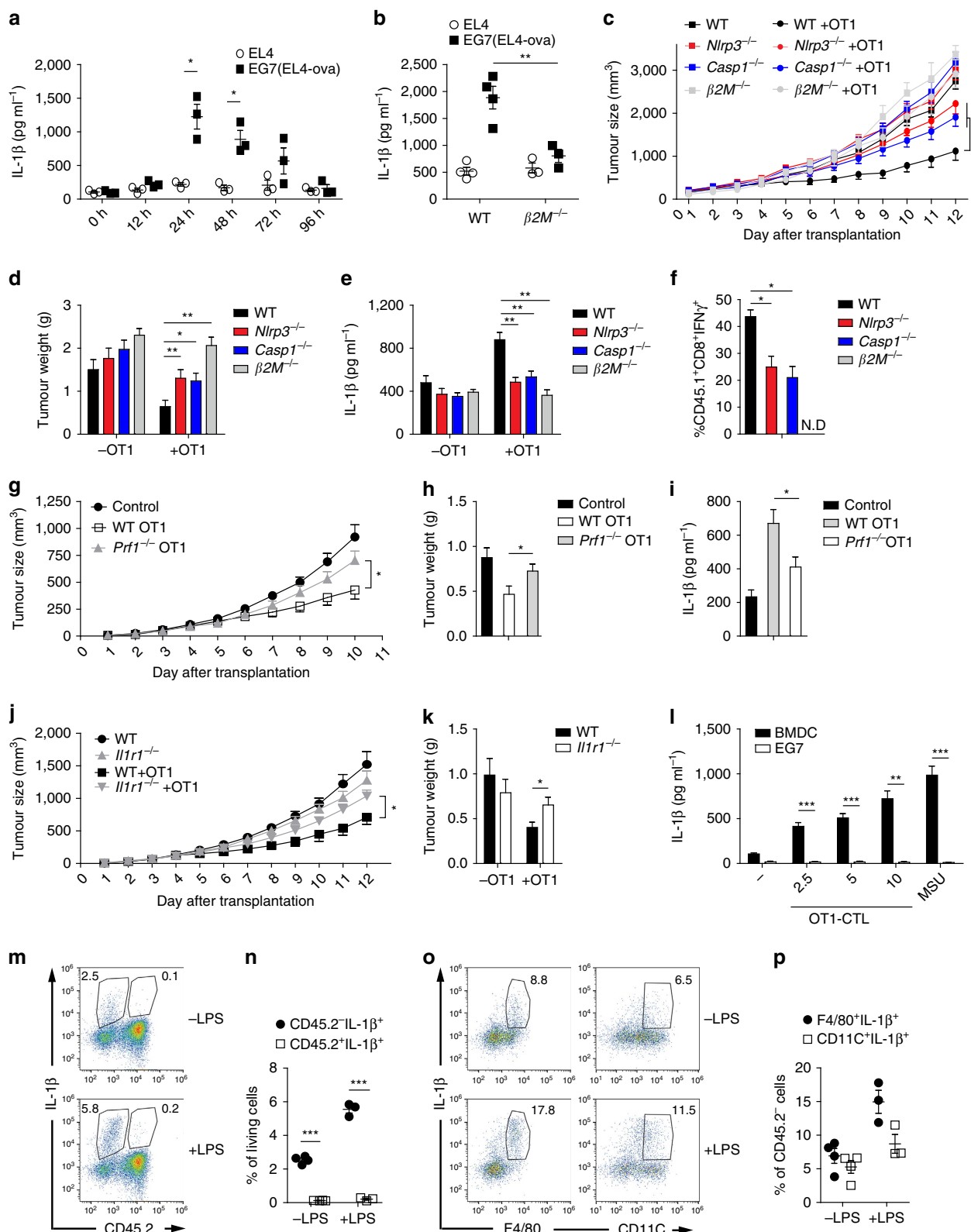

CTL-induced IL-1β secretion in BMDCs but not in EG7 cells *in vitro* (Fig. 5l). We then utilized CD45.1 background mice to prepare the tumour model by injecting EG7 cells s.c. (CD45.2 background) and then i.v. injecting OT1 CTLs (CD45.2 background) to analyse IL-1β-producing cells *in vivo*. Consistent with the *in vitro* data, IL-1β-producing cells were CD45.2⁻ (Fig. 5m,n; Supplementary Fig. 5m), indicating that EG7 tumour cells and OT1 CTLs do not produce IL-1β. Further analyses showed that both macrophage and dendritic cells produce IL-1β (Fig. 5o,p; Supplementary Fig. 5m). These data indicate that OVA antigen in EG7 tumour cells is presented by APCs to activate OT1 CTLs, and the OT1 CTLs then activate NLRP3 inflammasome for IL-1β secretion in APCs via a feedback mechanism.

**CTL-activated NLRP3 inflammasome in GVHD.** We showed above that the inflammasome was activated in the MLRs (Fig. 1g–k). To determine whether NLRP3 is responsible for the inflammasome mediated IL-1β maturation in MLRs, we utilized *Nlrp3*-deficient mice and found that NLRP3 deficiency in BMDCs markedly blocked IL-1β secretion but did not affect cell death in the MLR of BALB/c anti-C57/BL6 or C3H anti-C57/BL6 (Fig. 6a,b; Supplementary Fig. 6a,b). As MLR CTLs play a critical role in the pathogenesis of GVHD[43], we explored whether the NLRP3 inflammasome activation contributes to GVHD pathology. We established a CTL-mediated GVHD model by i.v. injection of C3H.SW background bone marrow (BM) with or without CTLs into irradiated C57BL/6 background recipient mice. Similar to the *in vitro* results in the MLR (Fig. 6b), IL-1β secretion was induced in the GVHD model (Fig. 6c), while deficiency of NLRP3, Caspase-1 or β2M reduced IL-1β induction (Fig. 6c), indicating that the MLR CTLs activate NLRP3 inflammasome in APCs for IL-1β maturation and secretion *in vivo*. Furthermore, the GVHD-associated body weight loss was significantly blocked in the *Nlrp3*- or *Caspase-1*-deficient mice (Fig. 6d). Deficiency of NLRP3 or Caspase-1 led to reduced GVHD mortality, similar to the effect of β2M deficiency (Fig. 6e). Consistent with the phenotypes of body weight change and mouse survival rate, histological examination also showed that deficiency of NLRP3, caspase-1 or β2M clearly suppressed lymphocyte infiltration and tissue damage in the liver and small intestine (Supplementary Fig. 6c,d). We showed above that perforin was essential for CTL-induced IL-1β secretion *in vitro* (Fig. 4h). To determine whether the observation happens *in vivo*, we transferred BM with wild-type or *Perforin*-deficient CTLs to bm1 mice to induce GVHD. We found deficiency of perforin in CTLs resulted in reduced IL-1β induction (Fig. 6f). Perforin deficiency also led to reduction of GVHD-associated body weight loss and mortality (Supplementary Fig. 6e,f), consistent with

previous reports[44]. Similar to the observation in the tumour model (Supplementary Fig. 5i,j), effector memory CTLs were the major population and had the highest expression of perforin and granzyme B in the target tissues of the GVHD model (Supplementary Fig. 6g,h), indicating their roles in GVHD pathogenesis. We next determined the potential role of MLR-CTL-induced IL-1β secretion in APCs in GVHD. We found that antibody blockade of IL-1β significantly reduced the CTL-mediated GVHD pathogenesis, including body weight change, survival (Fig. 6g,h) and pathology (Supplementary Fig. 6i,j). Furthermore, IL-1R1 deficiency significantly blocked pathogenesis in the GVHD model (Fig. 6i,j; Supplementary Fig. 6k,l). Consistently, CTL activation was reduced in the receptor-deficient mice in the GVHD model (Supplementary Fig. 6m). However, we found that IL-1β did not promote CTL activation and proliferation *in vitro* under the conditions with or without TCR activation (Supplementary Fig. 7). IL-1β has been reported to affect DC maturation and migration in a CCR7-dependent manner to prime CTLs[41]. We observed that the percentage of CCR7⁺ DC was reduced in the GVHD model of the receptor-deficient mice (Supplementary Fig. 6n). These results indicate that IL-1β does not directly target CTLs but instead activates other cell types, such as DCs, to prime CTLs.

**NLRP3 activation by CTL is not important for anti-infection.** In addition to antigen-specific antitumour immunity and GVHD, CTLs also have important functions in host defense against intracellular bacterial or viral infections. To determine whether intracellular bacterial antigen-specific CTLs can induce IL-1β secretion in APCs, we purified CTLs from wild-type mice challenged with OVA-containing *Listeria monocytogenes* (LM-OVA) and co-cultured the CTLs with BMDCs that were primed with LPS and pulsed with OVA peptide. We found that LM-OVA antigen-specific CTLs induced IL-1β secretion and cell death in a dose-dependent manner (Fig. 7a; Supplementary Fig. 8a). Similarly, we purified CTLs from mice challenged with LCMV and co-cultured the CTLs with BMDCs that were primed with LPS and pulsed with gp33, a peptide encoded by LCMV, and found that LCMV antigen-specific CTLs induced IL-β secretion in BMDCs (Fig. 7b; Supplementary Fig. 8b). These data suggest that antigen-specific CTLs from intracellular bacterial or viral infections can induce IL-1β secretion in APCs, similar to tumour antigen or MLR antigen-specific CTLs. To know whether the antigen-specific IL-1β induction in APCs by CTLs contributes to host defense against intracellular bacterial or viral infections, we utilized β2M-deficient mice because β2M-deficient mice are reported to have no functional CTLs and β2M was required for antigen-specific CTL-mediated IL-1β induction in APCs

**Figure 5 | CTL-induced NLRP3 activation in antitumour immunity.** (**a,b**) IL-1β levels in tumour homogenates from wild-type mice (**a**) and wild-type or *β2M⁻/⁻* mice (**b**) injected s.c. with EL4 or EG7 (EL4-ova) and then injected i.v. with activated OT1 CTLs for the indicated time (**a**) n = 3 or 24 h (**b**) n = 4 WT, n = 3 *β2M⁻/⁻*. (**c,d**) Tumour size (**c**) and tumour weight (**d**) from the indicated mice first injected s.c. with EG7 cells and then injected i.v. with OT1-CTLs. n = 11 WT, n = 7 *Nlrp3⁻/⁻*, n = 11 *Casp1⁻/⁻*, n = 5 *β2M⁻/⁻*. (**e**) IL-1β level in tumour homogenates at day 5 after OT1 CTLs injection from the mice treated as in **c,d**. n = 8 WT, n = 7 *Nlrp3⁻/⁻*, n = 5 *Casp1⁻/⁻*, n = 5 *β2M⁻/⁻*. (**f**) The T-cell percentage in tumours of the indicated mice first injected s.c. with EG7 cells and then injected i.v. with CD45.1⁺ OT1 CTLs. n = 5 WT, n = 7 *Nlrp3⁻/⁻*, n = 5 *Casp1⁻/⁻*, n = 5 *β2M⁻/⁻*. (**g–i**) Tumour size (**g**), tumour weight (**h**) and IL-1β level (**i**) in tumours from mice first injected s.c. with EG7 cells and then injected i.v. with OT1-CTLs or *Prf1⁻/⁻* OT1-CTLs. n = 8 per group. (**j,k**) Tumour size (**j**) and tumour weight (**k**) from wild-type and *Il1r1⁻/⁻* mice first injected s.c. with EG7 cells and then injected i.v. with OT1-CTLs. n = 8 per group. (**l**) IL-1β release from LPS-primed and ova-pulsed BMDCs or EG7 tumour cells co-cultured with OT1-CTLs. (**m,n**) Flow analysis (**m**) and quantification (**n**) of IL-1β producing cells in tumours from CD45.1⁺ mice first injected s.c. with EG7 cells (CD45.2⁺) and then injected i.v. with OT1-CTLs (CD45.2⁺) for 5 days. n = 4 − LPS, n = 3 + LPS. (**o,p**) Flow analysis (**o**) and quantification (**p**) of IL-1β producing cells gated from CD45.2⁻ cell population in **m**. n = 4 − LPS, n = 3 + LPS. (**a,b** and **c–k**, **m–p**) followed the procedures in Supplementary Fig. 5a and 5d respectively. Data are representative of three (**a–p**) independent experiments. Error bars, s.e.m. Two-way ANOVA (**c,g,j**), two-tailed Student's *t*-test (**a,b,d–f,h,i,k,l,n**), *P < 0.05, **P < 0.01, ***P < 0.001.

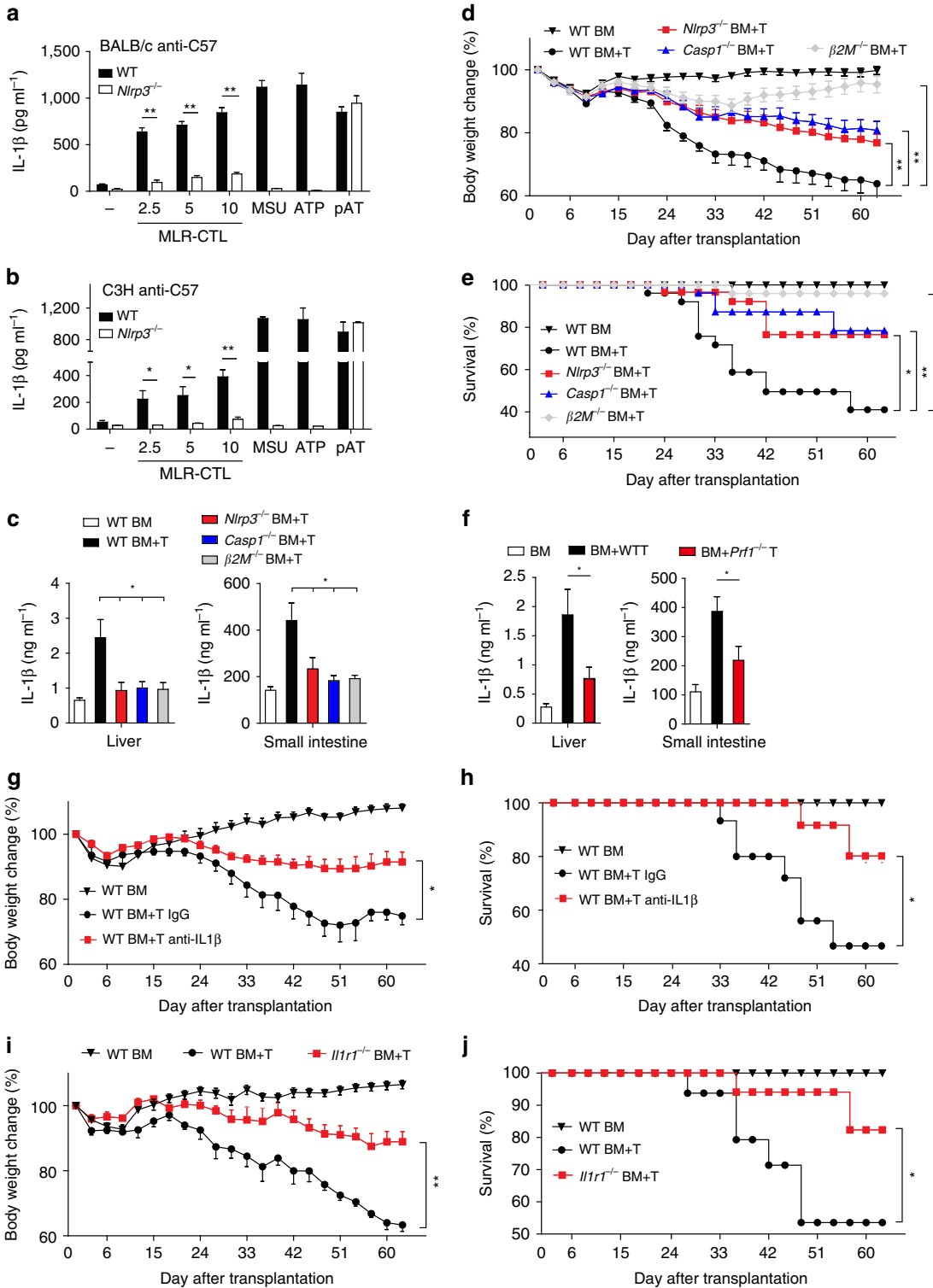

**Figure 6 | CTL-induced NLRP3 activation in GVHD.** (**a,b**) IL-1β release from LPS-primed wild-type or *Nlrp3*^−/− BMDCs co-cultured with BALB/c anti-C57 (**a**) or C3H anti-C57 (**b**) MLR-CTLs. (**c–e**) The level of IL-1β (**c**) n = 5 per group, body weight change (**d**) and survival rate (**e**) of wild-type, *Nlrp3*^−/−, *Casp1*^−/− or *β2M*^−/− mice underwent total body irradiation (TBI) followed by i.v. injection of C3H BM alone or C3H BM plus CTLs. n = 9 WT BM, n = 17 WT BM + T, n = 14 *Nlrp3*^−/− BM + T, n = 11 *Casp1*^−/− BM + T, n = 13 *β2M*^−/− BM + T. (**f**) The level of IL-1β in the liver or small intestine of the mice underwent TBI followed by i.v. injection of bm1 BM alone or bm1 BM plus CTLs from wide-type or *Prf1*^−/− mice. n = 6 per group. (**g,h**) Body weight change (**g**) and survival rate (**h**) of control IgG or anti-IL1β treated mice underwent TBI followed by i.v. injection of C3H BM alone or C3H BM plus CTLs. n = 5 WT BM, n = 8 WT BM + T IgG, n = 8 WT BM + T anti-IL1β. (**i,j**) Body weight change (**i**) and survival rate (**j**) of wild-type, *Il1r1*^−/− mice underwent TBI followed by i.v. injection of C3H BM alone or C3H BM plus CTLs. n = 5 WT BM, n = 8 WT BM + T, n = 8 *Il1r1*^−/− BM + T. Data are representative of three (**a–j**) independent experiments. Error bars, s.e.m. two-tailed Student's *t*-test (**a–c,f**), two-way ANOVA (**d,g,i**), log-rank test (**e,h,j**). *P < 0.05, **P < 0.01, ***P < 0.001.

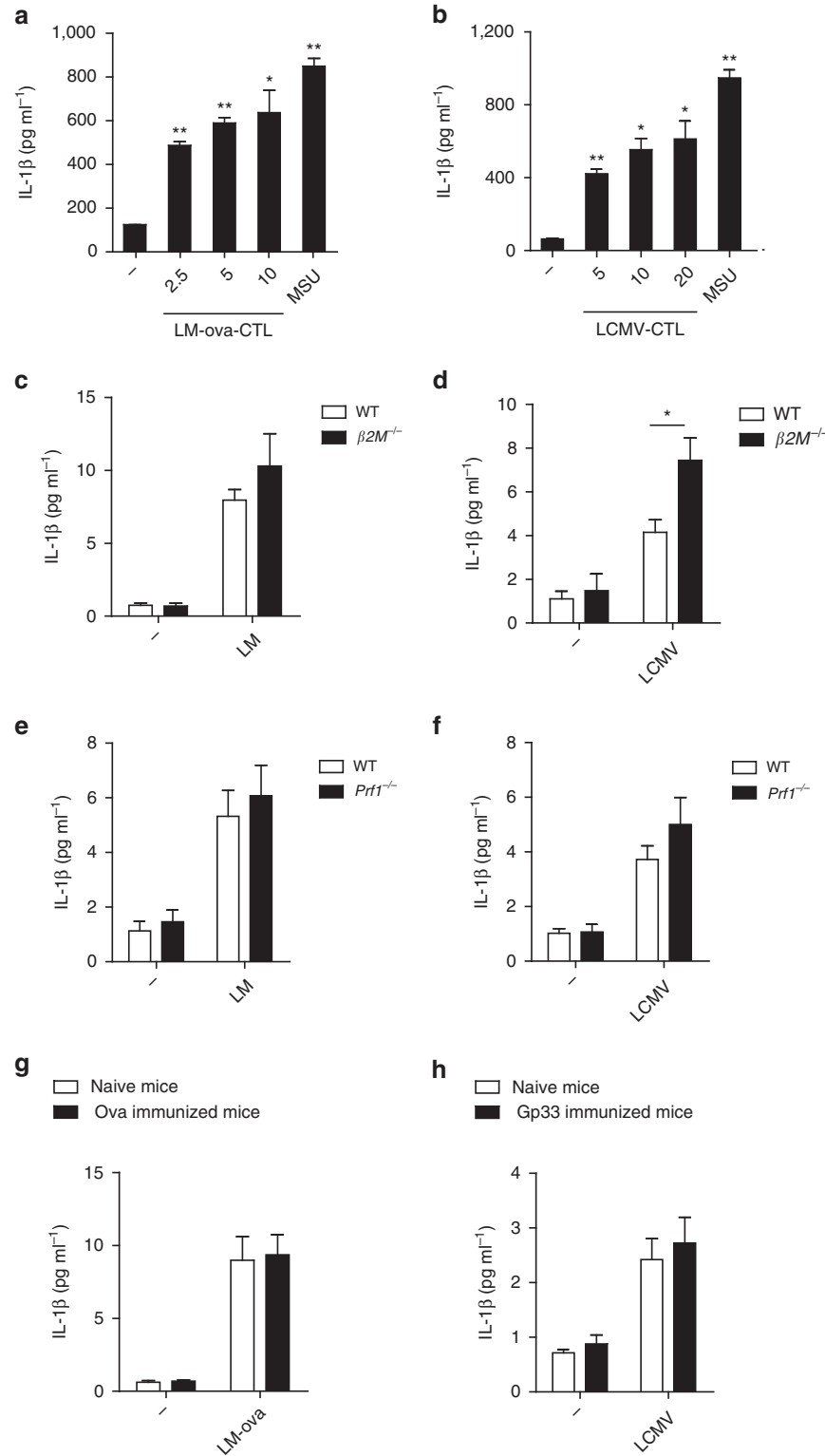

**Figure 7 | NLRP3 activation by CTL is not important for anti-LM or LCMV immunity.** (**a**) IL-1β secretion in ova-plused BMDCs after co-cultured with the CTLs from *Listeria monocytogenes*-ova immunized mice (LM-ova-CTL) at the indicated ratios for 4 h. (**b**) IL-1β secretion in gp33-plused BMDCs after co-cultured with the CTLs from LCMV immunized mice (LCMV-CTL) at the indicated ratio for 4 h. (**c**) IL-1β level in spleen from wild-type or $\beta 2M^{-/-}$ mice after i.v. injection of *Listeria monocytogenes* (LM). $n = 5$ per group. (**d**) IL-1β in spleen from wild-type or $\beta 2M^{-/-}$ mice after i.p. injection of LCMV virus. $n = 5$ per group. (**e**) IL-1β level in spleen from wild-type or $Prf1^{-/-}$ mice after i.v. injection of *Listeria monocytogenes* (LM). $n = 6$ per group. (**f**) IL-1β level in spleen from wild-type or $Prf1^{-/-}$ mice after i.p. injection of LCMV virus. $n = 6$ per group. (**g**) IL-1β level in spleen at day 5 after i.v. injection of LM-ova from the mice with or without ova pre-immunization for one month. $n = 5$ per group. (**h**) IL-1β level in spleen at day 5 after i.v. injection of LCMV from the mice with or without gp33 pre-immunization for one month. $n = 5$ per group. Data are representative of three (**a**–**h**) independent experiments. Error bars, s.e.m. *$P < 0.05$, **$P < 0.01$ by two-tailed Student's $t$-test.

(Fig. 3c,g). However, we found that IL-1β levels in the spleens of β2M-deficient mice were not reduced compared to wild-type control mice after either LM or LCMV infections (Fig. 7c,d). The increased IL-1β in β2M-deficient spleen (Fig. 7d) is probably due to increased viral titre that has been reported[45,46]. We also used the LM bacteria or LCMV virus to infect wild-type or *Perforin*-deficient mice. Similar to the observations in the *β2M*-deficient mice, we found that IL-1β levels were not reduced in the spleen of *Perforin*-deficient mice compared to wild-type mice in the infection models (Fig. 7e,f). We then checked into the potential effect of memory CTLs on IL-1β secretion *in vivo* by immunizing mice with OVA or gp33 peptides for one month and then challenging the mice with LM-OVA or LCMV infections respectively. There existed OVA antigen-specific CTLs in the OVA peptide immunized mice because the antigen-specific CTLs induced IL-1β secretion in BMDCs pulsed with OVA peptide *in vitro* (Supplementary Fig. 8c). However, OVA pre-immunization did not affect IL-1β level after LM-OVA challenge *in vivo* (Fig. 7g). Similarly, while the antigen-specific CTLs from gp33 peptide immunized mice induced IL-1β secretion in BMDCs *in vitro* (Supplementary Fig. 8d), gp33 pre-immunization did not affect IL-1β level after LCMV infection *in vivo* (Fig. 7h). These data suggest that antigen-specific CTL-induced IL-1β in APCs is not important for host defense against infections of virus and intracellular bacteria.

## Discussion

Here, we found that antigen-specific CTLs feedback promoted NLRP3 inflammasome activation in APCs for IL-1β maturation. By utilizing the MLR-CTL and OT1-CTL systems and *β2M* genetic-deficient mice, we demonstrated that CTL-mediated NLRP3 inflammasome activation required antigen presentation both *in vitro* and *in vivo*. As IL-1β is reported to play an important role in activating APCs to promote priming, proliferation and activation of CD4$^+$ T cells and CTLs[41,47], our study suggests that NLRP3 inflammasome activation by CTLs forms a positive feedback amplification loop to amplify antigen-specific adaptive immune responses. While the amplification loop is critical for antitumour immunity and GVHD pathogenesis, it is not important for host defense against either primary or secondary infections of virus and intracellular bacteria. The reason is likely because pathogens activate pathogen recognition receptors to induce very strong inflammasome activation for IL-1β secretion in APCs which overrides the effect of antigen-specific CTLs on IL-1β secretion[48]. Therefore, the feedback amplification loop is specifically needed for antigen-specific antitumour immunity or GVHD pathogenesis where innate immunity is probably not directly activated by tumour antigens or allo-antigens.

NLRP3 inflammasome can be activated by a variety of stimuli including pathogens and danger signals such as ATP and MSU[3,4,5,49–52]. We demonstrated that perforin in CTLs activated NLRP3 inflammasome in APCs in an antigen-specific manner. In contrast to the reported danger signals that are secreted from cells, we found that the CTL-mediated NLRP3 inflammasome activation requires cell–cell contact, consistent with perforin-mediated cell killing in a cell–cell contact-dependent manner[53]. We also provided evidence that apoptosis and necroptosis were not required for perforin-induced IL-1β secretion. NLRP3 has been reported to be activated by several common mechanisms, including calcium influx[37] and potassium efflux[39]. Perforin can form membrane pores in target cells and induce calcium influx[54]. We found that both calcium influx and potassium efflux were important for CTL-induced IL-1β secretion. Therefore, the perforin pore-forming ability likely induces calcium influx and potassium efflux to activate NLRP3 inflammasome.

Antigen-specific CTLs have important functions in host antitumour immunity and in the pathogenesis of GVHD. We found that the feedback activation of NLRP3 inflammasome in APCs by CTLs contributed to their killing effects. We established an antigen-specific antitumour mouse model through OT1-mediated killing of EG7 tumour cells containing OVA. The OT1-mediated killing of target tumour cells *in vivo* induced IL-1β production through antigen presentation and NLRP3 inflammasome, indicating that the OVA tumour antigen was presented by APCs to activate OT1 T cells, and the CTLs then activated NLRP3 inflammasome in APCs. Importantly, we found that CTL-mediated antitumour immunity required NLRP3 inflammasome. Although one recent study reported that mice with a genetic deficiency of NLRP3 inflammasome had increased colon cancer[12], the mechanisms of the enhanced cancer pathology are not known[55]. To our knowledge, our study provides the first evidence that NLRP3 inflammasome is required for antigen-specific immunity against tumours. Similarly, we showed that IL-1β was required for antigen-specific antitumour immunity, consistent with the reported antitumour function of IL-1β in several studies[14,56]. However, many studies reported the function of IL-1β in promoting tumourigenesis likely through its pro-inflammatory effects as chronic inflammation promotes cancer[57]. These data suggest that IL-1β has dual functions in tumour development, that is, antitumour immunity through tumour antigen-specific CTL-mediated effects and pro-tumour effects in a chronic inflammatory environment in an antigen-independent manner. Thus, targeting IL-1β for cancer therapy should consider its dual effects by differential strategies such as blocking the chronic inflammation-driven pro-tumour effect and strengthening antigen-specific antitumour immunity in a tumour-antigen vaccine, CAR-T therapy and immune checkpoint (CTLA-4 and PD-1) blockade. Similar to tumour antigen-specific CTL-mediated antitumour immunity, we found that MLR antigen-specific CTLs activated NLRP3 inflammasome for IL-1β production in APCs, and the CTL-mediated GVHD induced IL-1β production through NLRP3 inflammasome. While NLRP3 and IL-1β has been reported to be required for T-cell-(CD4$^+$ and CD8$^+$) mediated GVHD through pathological Th17 induction[13,58], our study showed that NLRP3 inflammasome is also required for CTL-mediated GVHD through promoting CTL functions. We have data to show that deficiency of perforin in CTLs resulted in significantly decrease in IL-1β level in GVHD target tissues as well as GVHD pathology, indicating that perforin is an important NLRP3 activator in GVHD model *in vivo*. Similarly in the antigen-specific antitumour model, we showed that deficiency of perforin in OT1 CTLs resulted in significantly reduced IL-1β level and consequently increased tumour size and weight. However, perforin can also mediate cell killing and thus may release factors like ATP to activate inflammasome secondarily or indirectly *in vivo*[59]. Together, our data indicate that antigen presentation by APCs activates CTLs, and the antigen-specific CTL feedback then activates NLRP3 inflammasome in APCs to form a positive feedback loop to amplify the CTL-mediated effects in antitumour immunity and GVHD pathogenesis.

Innate immunity instructs antigen-specific adaptive immunity. Here, we discovered an amplification loop of adaptive immunity to innate immunity through antigen-specific delivery of perforin to APCs to boost immune responses via NLRP3 inflammasome activation. Consistent with the perforin requirement for amplifying effective immunity, we observed that perforin expression was highly induced in effector memory CTLs (T$_{EM}$) compared to

naïve and central memory CTLs, and the $T_{EM}$ cells are the majority of the population of CTLs in models of both antitumour immunity and GVHD. Coincident with the perforin level, antigen-specific $T_{EM}$ cells induced much higher IL-1β levels in APCs than the other CTL populations. Our study indicates that innate immunity needs to be promoted during the initial or effector stage to enhance the antigen-specific antitumour immunity and GVHD responses. However, innate immunity might be deleterious during the resolution stage and may need to be suppressed for central memory formation. One potential mechanism is the suppression of adaptive immunity on innate immunity[22]. Treg cells have been reported to suppress both innate and adaptive immune responses either through cell–cell contact or through secreting suppressive cytokines, such as IL-10 and TGFβ, in an antigen-independent manner[23]. One study showed that anti-CD3-activated CD4$^+$ and CTLs suppressed NLRP3 inflammasome activation in macrophages *in vitro* in an antigen-independent manner[24]. We also confirmed these results that anti-CD3-activated CTLs inhibited NLRP3 inflammasome. In contrast, we unexpectedly found that antigen-specific CTLs promoted NLRP3 inflammasome in APCs. The reason for the differential antigen-dependent versus antigen-independent effects of CTLs is that antigen presentation is required for the cell–cell contact-dependent delivery of perforin from CTLs to APCs. Consistently, we found that perforin was required only for antigen-specific CTL-mediated promoting effects but was not important for antigen-independent CTL-mediated suppressive effect.

In summary, we demonstrate that antigen-specific CTLs activate NLRP3 inflammasome in APCs, and this is specifically required for the CTL-mediated effects in antitumour immunity and GVHD. We discover that perforin from antigen-specific CTLs is essential for the CTL-mediated innate NLRP3 inflammasome activation. Our study reveals a positive feedback loop of adaptive immunity to promote innate immunity to amplify antigen-specific adaptive immune responses in situations where innate immunity is not strongly activated.

## Methods

**Mice.** C57BL/6 (H-2$^b$), C3H.SW (H-2$^b$), Nlrp3$^{-/-}$ (C57BL/6 background), Casp1$^{-/-}$ (C57BL/6 background), Gzma$^{-/-}$Gzmb$^{-/-}$ (129X1/svj background), prf1$^{-/-}$ (C57BL/6 background), Aim2$^{-/-}$ (C57BL/6 background), Il1r1$^{-/-}$ (C57BL/6 background) and B6.C-H2bm1/ByJ (bm1) (C57BL/6 background) mice were purchased from the Jackson Laboratory. Rip3$^{-/-}$ (C57BL/6 background) mice was generated by Ripk3$^{tm1(KOMP)Wtsi}$ sperm purchased from KOMP Repository. CD45.1 wild-type (C57BL/6 background), H-2KbDb$^{-/-}$ (C57BL/6 background), β2M$^{-/-}$ (C57BL/6 background), 129X1/svj, Gzmb$^{-/-}$ (129X1/svj background) and Gzma$^{-/-}$ (129X1/svj background) mice were provided by Dr Yufang Shi (Institute of Health Sciences, Shanghai, China). OT1 TCR transgenic mice on C57BL/6 background specific for ova peptide (SIINFEKL) in the context of H-2K$^b$ were previously described[60]. Prf1$^{-/-}$ OT1 (C57BL/6 background) mice were generated by mating prf1$^{-/-}$ mice with OT1 mice in our laboratory. B6.MRL-lpr/lpr (H-2$^b$) (C57BL/6 background) mice were from model animal research center of Nanjing University. MRL-lpr/lpr (H-2$^k$) and MRL (H-2$^k$) control mice were provided by Dr Nan Shen (Institute of Health Sciences, Shanghai, China). BALB/c (H-2$^d$) mice were purchased from Shanghai Laboratory Animal Center, Chinese Academy of Sciences. All mice were maintained in specific pathogen-free conditions and age-matched littermate mice used for experimental purposes were between 6 and 10 weeks of age. All animal studies were performed in compliance with the guide for the care and use of laboratory animals and are approved by the Institutional Biomedical Research Ethics Committee of the Shanghai Institutes for Biological Sciences (Chinese Academy of Sciences).

**Reagents and cell lines.** Antibody to NLRP3 (Cryo-2) (dilution 1:1,000) was from AdipoGen. Antibody to human perforin (pf-344) (dilution 1:10) was from MABTECH. Antibody to caspase-3 (9662, dilution 1:1,000) was from Cell Signaling Technology (CST). Antibodies to ASC (sc-22514-R) (dilution 1:500) and mouse Caspase-1 (P10) (sc-514) (dilution 1:1,000) were from Santa Cruz Biotechnology, Inc. Antibodies to Flag (F7425) (dilution 1:1,000) or Caspase-11 (17D9) (dilution 1:1,000) were from Sigma-Aldrich (St. Louis, MO). Anti-hemagglutinin (HA) (MMS-101R) (dilution 1:2,000) was from Covance.

Mouse IL-1β antibody (AF-401-NA) (dilution 1:2,000), mouse IL-1β (cat. DY401), IL-6 (cat. DY406) and IFNγ (cat. DY485) ELISA kits were obtained from R&D Systems (Minneapolis, MN). Anti-CD8α (Ly-2) (cat. 130-049-401) and anti-CD4 (L3T4) (cat. 130-049-201) microbeads were from Miltenyi Biotec. CytoTox 96 Non-Radioactive Cytotoxicity Assay kit (cat. G1780) was from Promega. GM-CSF was from R&D Systems (415-ML-010) or peprotech (315-03). OVA (257–264) and LCMV gp33 peptide were from SBS Genetech Co., Ltd. Caspase 1 inhibitor (Z-YVAD-FMK, ALX-260-154) and purified native perforin were from enzo life sciences, and perforin buffer was prepared according to the manual. ProfectP1 (0041) was from targetingsystems. Recombinant granzyme B (G9278), Concanavalin A (conA) (C2010), Concanamycin A (CMA), phorbol 12-myristate 13-acetate (PMA) (P1585), LPS (L6529), ATP (A6419) and poly (deoxyadenylicthymidylic) acid sodium salt (dA:dT) (P0883) were from Sigma-Aldrich (St. Louis, MO). CFSE (C34554) and Lipofectamine RNAiMAX Reagent (13778030) were from Invitrogen Life Technologies. Fluo-4 (F14201) was from Invitrogen. MSU was prepared as previously described[61]. El4 and EG7 cells were provided by Dr Yufang Shi and maintained with 1640 containing 10% (vol/vol) FBS, penicillin (100 U ml$^{-1}$) and streptomycin (100 μg ml$^{-1}$), and EG7 culture medium was supplemented with 0.4 mg ml$^{-1}$ G418. The cell lines were routinely checked by PCR to ensure they are not infected with mycoplasma.

**Preparation of BMDCs and BMDMs.** BMDCs were generated as described previously[60]. Briefly, BM cells were flushed from the femurs and tibias of gene-deficient or wild-type mice and subsequently depleted of red cells with ammonium chloride. $1 \times 10^6$ ml$^{-1}$ BM cells were cultured in RPMI 1640 containing 10% (vol/vol) FBS, 100 U ml$^{-1}$ penicillin, 100 μg ml$^{-1}$ streptomycin, and 20 ng ml$^{-1}$ GM-CSF. Cells were cultured at 37 °C, and fresh medium was added every 2 or 3 days. Six days after the culture, the cells were determined by FACS detection of CD11c and MHC class II antigen, and collected for further experiments. BMDMs were prepared. Briefly, BM cells after removal of red blood cells were cultured at $3 \times 10^6$ cells per well in six-well plates or $1.8 \times 10^7$ cells per 10 cm disk in DMEM containing 10% (vol/vol) FBS, 100 U ml$^{-1}$ penicillin, 100 μg ml$^{-1}$ streptomycin, and 30% L929 conditioned medium. Fresh medium was added every 2 days. On day 6, cells were collected for further experiments.

**Isolation of mouse PMs.** Mice were treated with 2 ml 4% thioglycolate medium (FTG from BD Biosciences) per mouse by i.p. injection, and 3 days later cells were isolated by peritoneal cavity lavage with 10 ml of cold PBS. Cells were seeded in 96-well plates in DMEM culture medium. After overnight, the supernatant was discarded, and the adherent cells were regarded as PMs.

**Generation of CTLs *in vitro*.** To generate alloreactive CTL cells from MLR *in vitro*, H-2$^b$ responder splenocytes from gene-deficient and C57BL/6 mice ($8 \times 10^6$ ml$^{-1}$) were cultured together with irradiated (2,000–3,000 rad) stimulator splenocytes from BALB/c (H-2$^d$) mice ($4 \times 10^6$ ml$^{-1}$) at a ratio of 2:1 for 5–6 days or at the indicated days. Half of the cultured medium was replaced with fresh 1,640 medium and IL-2 (100 U ml$^{-1}$) was added on day three. The same procedure was followed for MLRs with other alloreactive CTLs from MHC mismatched mouse strains. To obtain ova peptide-specific CTLs, spleen cells from naive OT1 mice were stimulated *in vitro* in the presence of 10 nM ova peptide for 6 days. Fresh 1640 medium and IL-2 (100 U ml$^{-1}$) was added on day 3. CTL cells were isolated from splenic cell culture or MLR by MACS anti-CD8α microbeads according to the manufacturer's instruction (Miltenyi Biotec, Auburn, CA, USA). The purity of sorted cells in this study was consistently more than 95%. *L. monocytogenes* contained OVA- (LM-OVA) or LCMV-specific CTL cells (LM-OVA-CTL and LCMV-CTL) were isolated from spleen and lymph nodes of LM-OVA ($2 \times 10^4$ by i.v.) or LCMV ($5 \times 10^5$ by i.p.) infected mice at day 5–7 by MACS anti-CD8α microbeads (Miltenyi Biotec). Complete Freund's adjuvant plus OVA (100 μg) or gp33 (200 μg) antigen immunized the mice for one month, then OVA-CTLs or gp33-CTLs were isolated from spleen and lymph nodes of OVA or gp33 antigen immunized mice by MACS anti-CD8α microbeads (Miltenyi Biotec).

**ELISA.** The cytokine production was assessed with IL-1β, IL-6 and IFN-γ ELISA kits (R&D Systems) according to the manufacturer's instructions. A standard curve was generated using known amounts of the respective purified recombinant mouse cytokines.

**RNA isolation and real-time quantitative PCR.** Total RNA was extracted from cells with TRIzol reagent according to the manufacturer's instructions (Invitrogen). For cDNA synthesis, RNA was reverse-transcribed with a PrimeScript RT Reagent kit (TaKaRa), and cDNA was then amplified by real-time PCR with a SYBR Premix ExTaq kit (TaKaRa) on an AbiPrism 7900 HT cycler (Applied Biosystems). The expression of target genes was normalized to expression of housekeeping gene Rpl13a. The primer sequences are provided in Supplementary Table 1.

**Immunoprecipitation and immunoblot analysis.** The detailed protocols are followed as previously described[60]. To investigate oligomerization of ASC, the cell

lysates were centrifuged at 6,000 r.p.m. for 10 min at 4 °C. After the pellets were washed with 1 ml PBS and chemically cross-linked with 2 mM of DSS (Pierce) for 30 min at room temperature with rotation, the cross-linked pellets were centrifuged at 6,000 r.p.m. for 15 min and dissolved directly in SDS sample buffer before being subjected to immunoblotting.

**Flow cytometry and cell sorting.** The following antibodies were used for flow cytometry analysis: anti-IFNγ (XMG1.2, 1:50), anti-CD45.1 (A20, 1:100), anti-CD45.2 (104, 1:100), anti-CD44 (NJTEN3, 1:50), anti-CD44 (IM7, 1:100), anti-CD62l (MEL-14, 1:100), anti-F4/80 (BM8, 1:100), anti-CCR7 (4B12, 1:100), anti-CD4 (RM4-5, 1:200), anti-CD8α (53-6.7, 1:200), anti-H-2K$^b$ (AF6-88.5.5.3, 1:200), anti-Ly6G (RB6-8C5, 1:100), anti-MHC class II (I-A/I-E) (M5/114.15.2, 1:100) and anti-CD11c (N418, 1:100) were from eBioscience. Anti-CD103 (2E7, 1:100) was from Biolegend. Annexin V apoptosis Detection kit (556547) was from BD Pharmingen. For intracellular cytokine staining, Cytofix/Cytoperm plus kit (BD, cat. 55508) was used. For Foxp3 staining, Foxp3 staining buffer set (eBioscience, cat. 00-5523-00) was used. Multiple-colour flow cytometric analysis was performed using FACSAria (BD Biosciences; Franklin Lakes, NJ, USA). For FACS sorting, cells stained with FACS antibodies were sorted on BD FACSAria device. FlowJo software was used for data acquiring and analysis.

**CTL assay.** CTL killing assay was measured using a non-radioactive method based on release of lactate dehydrogenase (LDH) from target cells. BMDCs were primed with LPS for 3 h, and pulsed with the ova peptides (50 nM) for 1 h at 37 °C. After three times washing, $5 \times 10^4$ BMDCs in 100 μl were added to 100 μl of various numbers of effector cells that had been plated in 96-round well plates to obtain target: effector cell (CTL cells) ratios of 1:2.5, 1:5 or 1:10. After 4 h of incubation, the supernatant was collected to measure killing efficiency and cytokine analysis. The medium or the target cells alone were used as the low-level control (spontaneous LDH release). For the high-level control (maximum LDH release), lysate buffer was added to the target cells. The mixed cells were incubated for 4 h and assayed for LDH release using a cytotoxicity detection kit (LDH). The percentage of cell-mediated cytotoxicity was determined by the following equation: cytotoxicity (%) = [[(effector target cell mix − effector cell control) − low-level control]/(high-level control − low-level control)] × 100.

**Immunofluorescence.** Cells were cultured on glass in complete medium. For staining ASC speck, LPS-primed and ova-pulsed BMDCs on coverslips were washed twice with PBS, incubated with CTLs that were pre-labelled with Alexa Fluor 647-CD8α (eBioscience) and fixed for 15 min at room temperature with 4% formaldehyde in PBS, and then were washed three times with PBS. After permeabilization with 0.1% Triton X-100 in PBS and blocking with 5% BSA in PBS for 1 h, cells were incubated with anti-NLRP3 (Cryo-2; AdipoGen) and anti-ASC (sc-22514-R; Santa Cruz Biotechnology, Inc.) primary antibodies (in 5% BSA) for 1 h at room temperature. Cells were washed three times with PBS and then were incubated for 1 h at room temperature with the appropriate fluorescence-conjugated secondary antibody (Invitrogen). The samples were washed three times in PBS, and then incubated with DAPI for 5 min. Samples were observed with a Zeiss LSM510.

**Mouse peritoneal neutrophil recruitment model.** Mice were pre-injected with 10 μg ova peptide for 1.5 h, and then neutrophil influx was induced by injection i.p. of $2 \times 10^7$ CTL cells in 0.3 ml sterile PBS. After 6 h, mice were killed and peritoneal cavities were washed with 10 ml of PBS. The lavage fluids were analysed for neutrophils recruitment by FACS using the neutrophil marker CD11b-FITC (M1/70, 1:100) and Ly6G-PE (RB6-8C5, 1:100).

**Tumour graft model.** Mice received s.c. injections of $1 \times 10^6$ tumour cells in 100 μl of PBS in the right flank. Tumour size was measured at the indicated time points with a vernier caliper, and tumour size was estimated with the following formula[62,63]: width × width × length × 0.5. Raw data and tumour size calculation result are included as Supplementary Data 1. The mice were euthanized by CO$_2$ asphyxiation when they had tumour size over the maximum size 4,000 mm$^3$ or whenever they were observed in severe sick. Five to seven days after inoculation of the tumour cells, the mice were adoptively transferred i.v. with $1.5 \times 10^7$ OT1 CTLs that were isolated and purified by MACS anti-CD8α microbeads. Tumours were then isolated from the treated mice at the indicated times.

**GVHD induction model.** Allogeneic lethally irradiated mice underwent allogeneic BM transplantation as previously described[64]. Briefly, recipient mice were irradiated with 9.5 Gy administered in three fractions of 137Cs source. C3H.SW CD8$^+$ T cells ($2 \times 10^6$) plus C3H.SW TCD BM ($5 \times 10^6$) or C3H.SW TCD BM ($5 \times 10^6$) alone were i.v. given into the irradiated recipient mice. The bm1 mice possess a mutant class I allele that differs from B6 mice. Bm1 mice were irradiated with 5 Gy total body irradiation (TBI) and injected with $2 \times 10^6$ CD8$^+$ T cells together with $5 \times 10^6$ TCD BM cells from donors. T cells were isolated from the donor spleen with the MACS cell separation system (Miltenyi Biotec, Germany).

Day of BM or BM plus CD8$^+$ T-cell injection was assigned as day 0. Recipient mice were monitored for survival and body weight two to three times a week. The levels of IL-1β in tissue homogenates were analysed by ELISA on day 7. Slides of small intestine and liver samples collected on day 35 were stained with H&E. The tissue was examined using a semiquantitative scoring system known to be associated with GVHD[65]. Seven parameters were scored for small intestine (villous blunting, crypt regeneration, crypt epithelial cell apoptosis, crypt loss, luminal sloughing of cellular debris, lamina propria inflammatory cell infiltrate and mucosal ulceration) and ten parameters for liver (portal tract expansion by an inflammatory cell infiltrate, lymphocytic infiltrate of bile ducts, bile duct epithelial cell apoptosis, bile duct epithelial cell sloughing, vascular endothelialitis, parenchymal apoptosis, parenchymal microabscesses, parenchymal mitotic figures, hepatocellular cholestasis and hepatocellular steatosis). The scoring system for each parameter was as follows: 0 indicates normal; 0.5, focal and rare; 1, focal and mild; 2, diffuse and mild; 3, diffuse and moderate; and 4, diffuse and severe.

**L. monocytogenes and LCMV infection.** L. monocytogenes strain 10403s (LM) or was prepared by shaking the bacteria overnight at 37 °C in brain heart infusion broth with 50 μg ml$^{-1}$ streptomycin. For infections, LM were grown to mid-logarithmic phase, then resuspended in PBS and quantified by visible spectrometry readings at 600 nm. LCMV Armstrong strain was provided by Dr Lilin Ye. LM ($2 \times 10^4$ by i.v.) or LCMV ($5 \times 10^5$ by i.p.) infected mice for 5–7 day, then the levels of IL-1β in tissue homogenates were analysed by ELISA.

**Neutralization with IL-1β antibody.** The neutralizing InVivoMAb anti-mouse IL-1β monoclonal antibody (mAb) (Catalogue#: BE0246) and normal InVivoMAb Polyclonal Armenian Hamster IgG (Catalogue#: BE0091) were purchased from BioXCell. In the GVHD induction model, a total of 400 μg of anti-IL-1β mAb or control IgG was administered intraperitoneally into mice at day − 1 day before TBI and every 5 day after TBI.

**Isolation of LP lymphocytes.** Mice were killed and intestines were washed with Hank's balanced salt solution (HBSS) buffer after dissection of fat and mesenteric tissue and excision of Peyer's patches. After intestines were cut into 1 cm pieces longitudinally and washed with HBSS buffer, epithelia were removed by 250 r.p.m. shacking at 37 °C in HBSS buffer containing 30 mM EDTA, 1 mM DTT and 5% FBS for 30 min. After sedimentation, the tissues were then digested in RPMI 1640 medium (Invitrogen) containing DNase I (Sigma) (150 μg ml$^{-1}$), collagenase VIII (Sigma) (200 U ml$^{-1}$), 5% (vol/vol) FBS, penicillin (100 U ml$^{-1}$) and streptomycin (100 mg ml$^{-1}$) at 37 °C in 5% CO2 incubator for 1–1.5 h. The digested tissues were homogenized by vigorous shaking and passed through 70 μm cell strainer (Falcon). Then, the supernatant was centrifuged at 1,200 r.p.m. for 5 min for collecting cells and the lamina propria lymphocytes were further isolated by Percoll (40%/80%; GE Healthcare).

**Statistics.** Statistical analyses were performed with the t-test for two groups or one-way ANOVA for multiple groups using GraphPad Prism (GraphPad Software). Data derived from the animals at several times were analysed with two-way ANOVA to evaluate differences between experimental groups. Differences in animal survival were analysed by log-rank test. P values of < 0.05 were considered significant. *P < 0.05, **P < 0.01, ***P < 0.001.

**Data availability.** The authors declare that the data supporting the findings of this study are available within the article and its Supplementary Information Files, or from the corresponding authors upon reasonable request.

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

## Acknowledgements

This work was supported by grants from the National Natural Science Foundation of China (81430036, 81230075, 91429307, 91329301, 91542119, 31329002 and 31400756), the 973 Program (2013CB944904), the Strategic Priority Research Program of the

Chinese Academy of Sciences (Grant No. XDB19000000) and China Postdoctoral Science Foundation funded project (Project No. 2014M551471).

## Author contributions

Y.Q. and Y.Y. designed the experiments and wrote the manuscript; Y.Y. conducted the experiments and analysed the data. S.C., M.C., X.F., T.Y., Y.H., X.S. and Y.L. helped with experiments. F.W., L.Y., Y.S., N.S. and X.L. provided reagents and technical support. Y.Q. supervised the study.

## Additional information

**Competing interests:** The authors declare no competing financial interests.

