## [Peer Review File · Nature Communications]

Reviewers' comments:

Reviewer #1 (Remarks to the Author):

Qian and colleagues report that CD8+ T cells feedback promote NLRP3 inflammasome activation in APCs for IL-1beta production. The authors show that Perforin of antigen-specific CTLs is essential for NLRP3 inflammasome activation in APCs. In a GVD model the authors show that the NLRP3 inflammasome activated by antigen-specific CD8+ CTLs is critical for disease pathogenesis.

Major points:

1. How do the authors define alloantigen-specific CD8+ CTLs from the MLR (Fig. 1i,j)?

Only a small fraction of the T cells exposed to the allogeneic DC will become alloantigen specific. These T cells need to be isolated (e.g. the highly proliferative T cells by CFSE dilution) and used then. Otherwise the cells cannot be named alloantigen specific.

2. Have the authors used Perforin alone to study if the molecule by itself can activate the NLRP3 inflammasome?

3. Figure 6: The data are convincing but comparable findings were already shown before in Jankovic et al. JEM 2013.

4. The finding of the authors that antibody blockade of IL-1β significantly reduced the CTL-mediated GVHD pathogenesis, including body weight change, survival (Fig. 6f,g) and pathology (Supplementary Fig. 6g,h) is consistent with several previous publications showing that blockade of IL-1β reduces GVHD (Jankovic et al. JEM 2013, Park MJ et al. Mediators Inflamm. 2015;2015:631384).

5. Others have previously shown that Granzyme A produces bioactive IL-1β through a nonapoptotic inflammasome-independent pathway: Hildebrand D et al. Cell Rep. 2014 Nov 6;9(3):910-7

6. During GVHD there are many other activators of the NLRP3 inflammasome present, e.g. ATP (Wilhelm K et al. Nature Medicine 2010). Can the authors discuss or rule out that in their model also AT released by the CTL plays a role besides perforin?

7. Perforin is well known to kill cells. Do the APCs really just release the IL-1beta or do they die and all intracellular content increases unspecifically?

Minor points:

1) the authors are inconsistent in the references: e.g.

in Ref 12. Allen, I.C. et al. they write Journal of Experimental Medicine while in Ref 13. Jankovic, D. et al. they write J Exp748 Med

also some references are confusion because the authors are not named but just initials: e.g. 16. Zijlstra M, B.M.S.N.E. & et al.

Reviewer #2 (Remarks to the Author):

This ground breaking study reports a new paradigm for CD8+ CTLs and perforin in activating the NLRP3 inflammasome in APCs, and demonstrates its importance in anti-tumour immunity and GVHD models, but not anti-microbial responses. The conclusions are well supported by an abundance of mechanistic in vitro and in vivo data that is derived from numerous, and relevant, gene targeted mice. Given the extensive nature of the manuscript as it stands, the enormous amount of supporting data, and the novelty and impact the findings will have in the field, I recommend that this study be

published without further experimental work.

Reviewer #3 (Remarks to the Author):

This paper nicely describes a phenomenon in which CD8 T cells elicit IL-1b secretion from APC upon antigen recognition. This is an interesting observation and the data provided in support of this conclusion is clear. Some additional controls are still necessary to fully support this argument though.

- Fig. 2J. What happens without the addition of OVA?
- Fig. 4g. The response to MSU is reduced upon treatment with NH₄Cl. Can the authors exclude a toxic/non-specific effect, when MSU should be able to activate the inflammasome regardless of NH₄Cl?
- Viability controls are needed for Fig. 4j+k.
- Inflammasome activation typically also leads to pyroptosis. Did the author measure cell death of the APC? Also, the consequences of such APC killing should be discussed in the context of a sustained antigen-specific T cell response.

What is less clear in this study is whether this phenomenon is causally linked to for the observed differences in the tumor and the GvHD model. I have the following queries/issues.

- The description of the experiments was incomplete and thus made it difficult to follow. For example, in the experiments in Fig. 5a, did the authors inject naïve or effector OT-I? This is a critical point, as priming of naïve OT-I and their subsequent migration to tumors would be inconsistent with the kinetics presented.
 - The link between changes in the IL-1b responses and reduced IFN-g responses (Fig. 5f) is not clear. Is the reduction in IFN-g due differences in antigen load or does the IL-1b feedback onto the OT-I modulate the IFN-g response?
- The authors state that all experiments were performed in accordance with their institutional ethics requirements. It is important to note that the tumors were allowed to grow to very large sizes (up to ~7000mm³). With most other studies required to cull mice at much smaller tumor sizes, it is difficult to put the results into perspective. In fact, when looking at tumors sizes smaller than 2000mm³, which is well beyond the end point of the majority of studies, the differences between the experimental conditions appear much less pronounced (Fig. 5c, g).
 - In Fig. 5b, the tumors should still express MHC-I molecules in the b2M^{-/-} mice. So unless, the authors injected naïve OT-I and thus required antigen presentation by the host (see my point on the kinetics above), it is not clear why antigen recognition on the tumor cells doesn't induce IL-1b.
 - The first paragraph of the discussion needs to better reflect that the described phenomenon played no role in the infectious models tested here and therefore is not a generalizable principle.

Overall, I found this study well conducted and very interesting, although I am still somewhat skeptical about how well the observations in the in vivo tumor/GvHD models reflect CTL-induced IL-1b production by APC upon antigen recognition.

REVIEWERS' COMMENTS:

Reviewer #1 (Remarks to the Author):

The authors have responded well to my comments. I have no further questions.

Reviewer #3 (Remarks to the Author):

The authors have addressed my concerns sufficiently. I have to say though that the withdrawal of the NH₄Cl data and the obviously grave mistakes in calculating tumor size are disappointing. I was more enthusiastic about the work before.

Dear Editor

Enclosed please find our revised manuscript entitled "Antigen specific CD8⁺ T cells feedback activate NLRP3 inflammasome in antigen presenting cells through perforin" (NCOMMS-16-22387). We thank all the reviewers for the insightful and positive comments. We have now carefully revised the manuscript according to all the comments from the reviewers. We appreciate the opportunity you offered us to revise our manuscript. The followings are our point-by-point responses.

Reviewers' comments:

Reviewer #1 (Remarks to the Author):

Qian and colleagues report that CD8⁺ T cells feedback promote NLRP3 inflammasome activation in APCs for IL-1beta production. The authors show that Perforin of antigen-specific CTLs is essential for NLRP3 inflammasome activation in APCs. In a GVD model the authors show that the NLRP3 inflammasome activated by antigen-specific CD8⁺ CTLs is critical for disease pathogenesis.

We thank the reviewer for the positive comments and the points raised to improve our manuscript.

Major points:

1. How do the authors define alloantigen-specific CD8⁺ CTLs from the MLR (Fig. 1i,j)?

Only a small fraction of the T cells exposed to the allogeneic DC will become alloantigen specific. These T cells need to be isolated (e.g. the highly proliferative T cells by CFSE dilution) and used then. Otherwise the cells cannot be named alloantigen specific.

The point is well taken. According to your suggestion, we isolated CFSE^{high} CD8⁺ CTLs from MLR as non-responding cells and CFSE^{low} CD8⁺ CTLs from MLR as alloantigen specific CD8⁺ CTLs¹, then co-cultured them with allogeneic DC cells and measured IL-1 β secretion. We found that CFSE^{low} CD8⁺ CTLs (alloantigen specific CD8⁺ CTLs) induced IL-1 β secretion after co-culture as indicated in Fig. 1l and Supplementary Fig. 1f.

2. Have the authors used Perforin alone to study if the molecule by itself can activate the NLRP3 inflammasome?

The point is well taken. Perforin needs to be dissolved in calcium containing buffer when added to culture medium to treat APCs, and calcium is reported to be important for the function of perforin^{2, 3, 4}. We found that IL-1 β secretion was already highly induced in the supernatant of BMDCs that were treated with only perforin buffer (Fig. 4j), likely due to the high extracellular calcium concentration in the buffer because calcium is reported to be essential for NLRP3 activation⁵ and the calcium inhibitor or NLRP3 deficiency blocked the buffer induced IL-1 β secretion (Fig. 4j). Under this condition, perforin still statistically increased IL-1 β secretion although the increase was not dramatic (Fig. 4j), and the increase of IL-1 β secretion by perforin was blocked by the calcium inhibitor or NLRP3 deficiency (Fig. 4j). We also utilized the profectP1 protein delivery system and found that perforin delivered to the target cells (BMDCs) dramatically induced IL-1 β secretion and NLRP3 was required for perforin-induced IL-1 β secretion under this delivery system (Fig. 4k).

3. Figure 6: The data are convincing but comparable findings were already shown before in Jankovic et al. JEM 2013.

We are sorry for not describing our point clearly to show Figure 6. The point here is to use GVHD model to support our *in vitro* data that antigen specific CTLs activate NLRP3 inflammasome. Although the phenotype in our GVHD model is similar to the one in Jankovic et al. JEM 2013 which we cited in the previous submission, the proposed mechanisms are different. The GVHD model in the JEM paper utilized CD8⁺/CD4⁺ T cells and NLRP3-IL-1 β axis was proposed to exert its effects to shape Th17 responses to contribute to GVHD pathology. However, we only transferred CTLs (without CD4⁺ T cells) in our GVHD model and used the model to demonstrate that alloantigen specific CTLs induce IL-1 β secretion through NLRP3 inflammasome and IL-1 β feedback promotes DC priming and CTL activation. We now have data to show that perforin in CTLs contributed to IL-1 β secretion and consequently to CTL-mediated GVHD pathogenesis (Fig. 6f and Supplementary Fig. 6e,f).

4. The finding of the authors that antibody blockade of IL-1 β significantly reduced the CTL-mediated GVHD pathogenesis, including body weight change, survival (Fig. 6f,g) and pathology (Supplementary Fig. 6g,h) is consistent with several previous publications showing that blockade of IL-1 β reduces GVHD (Jankovic et al. JEM 2013, Park MJ et al. Mediators Inflamm. 2015;2015:631384).

We cited the JEM paper in our previous submission and have now cited the other paper. As described above, we utilized CTLs without CD4⁺ T cells in our GVHD model to demonstrate that alloantigen specific CTLs induce IL-1 β secretion through NLRP3 inflammasome and IL-1 β feedback promotes DC priming and CTL activation. Both “Jankovic et al. JEM 2013” and “Park MJ et al. Mediators Inflamm. 2015;2015:631384” report that blockade of IL-1 β signaling alleviates GVHD severity through suppressing Th17 cell differentiation. In our GVHD model without involvement of CD4⁺ T cells (i.e. no Th17 cells), we found blockade of IL-1 β signaling alleviated GVHD severity through suppression CTL activation.

5. Others have previously shown that Granzyme A produces bioactive IL-1 β through a nonapoptotic inflammasome-independent pathway: Hildebrand D et al. Cell Rep. 2014 Nov 6;9(3):910-7

This is a good point. We have now cited this paper. The paper has shown that *Pasteurella multocida* toxin (PMT) induces granzyme A expression in macrophage and the exocytosed granzyme A enters target cells and mediates IL-1 β maturation independently of caspase-1 and without inducing cytotoxicity. However, in our genetic mouse model system, we found that granzyme A in CTLs was not important for MLR-CTL-induced IL-1 β secretion by comparing granzyme A deficient CTLs with wild-type (WT) CTLs or comparing granzyme A and B double deficient CTLs with granzyme B deficient CTLs (Fig. 4h). Furthermore, we found that caspase-1 was required for IL-1 β maturation and secretion in APCs by MLR-CTLs or OT-1 CTLs (Fig. 2e,f,i, Fig. 5e and Fig. 6c). In addition, the expression of perforin and granzymes was very weak in BMDCs (Supplementary Fig. 4a) and deficiency of perforin, granzyme B or granzymes A and B in BMDCs did not affect the specific CD8⁺ CTL mediated induction of IL-1 β and cell death (Supplementary Fig. 4f). Thus the differential use of granzyme A and caspase-1 in IL-1 β maturation is probably due to the different experimental systems.

6. During GVHD there are many other activators of the NLRP3 inflammasome present, e.g. ATP (Wilhelm K et al. Nature Medicine 2010). Can the authors discuss or rule out that in their model also AT released by the CTL plays a role besides perforin?

The point is well taken. We have now cited the paper. As we described above, we utilized CD8⁺ T cells (without CD4⁺ T cells) in our GVHD model. We now have data to show that deficiency of perforin in CD8⁺ T cells resulted in significantly decrease in IL-1 β level in GVHD target tissues (liver and intestine) as well as GVHD pathology (body weight and survival) (Fig. 6f and Supplementary Fig. 6e,f), indicating that perforin is an important NLRP3 activator in our GVHD model. However, perforin can also mediate cell killing and thus may release factors like ATP etc. to activate inflammasome secondarily or indirectly in vivo. We have added this point discussion in the discussion section.

7. Perforin is well known to kill cells. Do the APCs really just release the IL-1beta or do they die and all intracellular content increases unspecifically?

We are sorry for not describing this point clearly. APCs are normally primed with agents like

LPS to induce pro-IL-1 β expression and inflammasome activation results in caspase-1 activation which cleaves pro-IL-1 β to mature IL-1 β . The mature IL-1 β is secreted through an unclear mechanism after inflammasome activation. In our system, we showed that antigen specific CTLs activated NLRP3 inflammasome including NLRP3-ASC association and ASC speck, and consequently caspase-1 activation for pro-IL-1 β cleavage and mature IL-1 β secretion by confocal, western blots and ELISA (Figure 1 – Figure 3). Without NLRP3 activation or caspase-1 activation, pro-IL-1 β could not be processed to mature IL-1 β and there was no increase in mature IL-1 β secretion although antigen specific CTLs still mediated target cell killing (Fig. 2 and supplementary Fig. 2). We also performed kinetic comparison of IL-1 β secretion with cell death (apoptosis and necrosis). IL-1 β release and apoptosis started at 1hr while cell necrosis (PI/Annexin-V double positive) at 2hr. We further found that apoptosis did not affect CTL-induced IL-1 β secretion. Collectively, our data suggest that IL-1 β release from APCs are due to the inflammasome activation but not just intracellular component release because of CTL-mediated target cell killing.

Minor points:

1) the authors are inconsistent in the references: e.g. in Ref 12. Allen, I.C. et al. they write Journal of Experimental Medicine while in Ref 13. Jankovic, D. et al. they write J Exp748 Med

The point is well taken. The references in the previous submission were automatically added by the endnote, and now we have manually corrected them.

also some references are confusing because the authors are not named but just initials: e.g. 16.

Zijlstra M, B.M.S.N.E. & et al.

The point is well taken. The references in the previous submission were automatically added by the endnote, and now we have manually corrected them.

Reviewer #2 (Remarks to the Author):

This ground breaking study reports a new paradigm for CD8+ CTLs and perforin in activating the NLRP3 inflammasome in APCs, and demonstrates its importance in anti-tumour immunity and GVHD models, but not anti-microbial responses. The conclusions are well supported by an abundance of mechanistic in vitro and in vivo data that is derived from numerous, and relevant, gene targeted mice. Given the extensive nature of the manuscript as it stands, the enormous amount of supporting data, and the novelty and impact the findings will have in the field, I recommend that this study be published without further experimental work.

We thank the reviewer for the very positive comments and for the recognition of large amount of data presented.

Reviewer #3 (Remarks to the Author):

This paper nicely describes a phenomenon in which CD8 T cells elicit IL-1 β secretion from APC upon antigen recognition. This is an interesting observation and the data provided in support of this conclusion is clear. Some additional controls are still necessary to fully support this argument though.

We thank the reviewer for the positive comments and the points raised to improve our manuscript.

- Fig. 2J. What happens without the addition of OVA?

The point is well taken. We have now added the control (Fig. 2j, bottom panel).

- Fig. 4g. The response to MSU is reduced upon treatment with NH₄Cl. Can the authors exclude a toxic/non-specific effect, when MSU should be able to activate the inflammasome regardless of NH₄Cl?

NH₄Cl and CMA are reported to be perforin inhibitors and were utilized in our study to support perforin is crucial for antigen specific CTL-mediated IL-1 β secretion. Lysosome is reported to play an important role in crystal-mediated NLRP3 activation. Bafilomycin A and NH₄Cl are reported to block the formation of acidic lysosome in cells^{6, 7}. Bafilomycin A is reported to inhibit NLRP3 activation in response to crystal like silia^{8, 9, 10}. We think NH₄Cl may also inhibit NLRP3 activation in response to MSU crystal through its effect on lysosome. However, we can not exclude its potential toxic or non specific effect on inflammasome activation by MSU. Therefore we removed the unnecessary data with NH₄Cl treatment and kept the data with CMA treatment in the panel (Fig. 4g). Actually we also have genetic evidence to support that perforin is required for antigen specific CTL-mediated IL-1 β secretion (Fig. 4h).

- Viability controls are needed for Fig. 4j+k.

The point is well taken. We have now added the controls (Supplementary Fig. 4h,i).

- Inflammasome activation typically also leads to pyroptosis. Did the author measure cell death of the APC? Also, the consequences of such APC killing should be discussed in the context of a sustained antigen-specific T cell response.

As mentioned above, we measured kinetics of cell death and IL-1 β release (Fig. 1e and Supplementary Fig. 3c) and found that IL-1 β release started before cell death determined by flow gating APCs with PI/Annexin-V double positive staining (Fig. 1e and Supplementary Fig. 3c). We agree with the reviewer that NLRP3 inflammasome activation typically also leads to pyroptosis in addition to IL-1 β maturation and release. Indeed, we showed that NLRP3 was required for ATP induced pyroptosis/cell death determined by LDH release (Supplementary Fig. 2a and Supplementary Fig. 6a,b). However, NLRP3 was not essential for MLR-CTL or OT1-CTL induced cell death (Supplementary Fig. 2a and Supplementary Fig. 6a,b), indicating that the cell death effect by the CTL-mediated target cell killing overrides the death effect by NLRP3 mediated pyroptosis. We also excluded caspase-11 mediated pyroptosis by using 129 mice with caspase-11 mutation in vitro (Supplementary Fig. 2c).

As described above, during sustained antigen-specific T cell response, perforin can mediate APC killing, resulting in release of DAMP factors like ATP etc. out of APCs to further activate inflammasome secondarily or indirectly in vivo in the models of anti-tumor and GVHD. We have added this discussion in the discussion section.

What is less clear in this study is whether this phenomenon is causally linked to for the observed differences in the tumor and the GvHD model. I have the following queries/issues.

- The description of the experiments was incomplete and thus made it difficult to follow. For example, in the experiments in Fig. 5a, did the authors inject naïve or effector OT-I? This is a critical point, as priming of naïve OT-I and their subsequent migration to tumors would be inconsistent with the kinetics presented.

We are sorry for not describing this point clearly. We used effector OT-I CTLs as described in Supplementary Fig. 5a. We have now described it more clearly in the figure legend.

- The link between changes in the IL-1 β responses and reduced IFN-g responses (Fig. 5f) is not clear. Is the reduction in IFN-g due differences in antigen load or does the IL-1 β feedback onto the OT-I modulate the IFN-g response?

The point is well taken. We compared the expression level of H2-K^b, the MHC I that specifically load ova peptide antigen on DC cells from WT, *Nlrp3*^{-/-}, *Casp1*^{-/-} and $\beta 2m$ ^{-/-} mice in the tumor model. We found DC cells from WT, *Nlrp3*^{-/-} and *Casp1*^{-/-} mice had a comparable MHC I level while $\beta 2m$ deficiency reduced MHC I expression (Supplementary Fig. 5h), indicating that the reduction of the number of IFN-g producing CTLs of NLRP3 or caspase-1 deficiency is likely not due to the differences in antigen load of DC cells while the reduction of the number of IFN-g producing CTLs of $\beta 2m$ deficiency is due to the defect in antigen load of DC cells. The reduction of the number of IFN-g producing CTLs in *Nlrp3*^{-/-} and *Casp1*^{-/-} mice are likely due to reduction of IL-1 β secretion because we showed that the number of IFN-g producing CTLs was reduced in IL-1R KO mice (Supplementary Fig. 5k). In the discussion, we mentioned that IL-1 β did not directly promote CTL activation and proliferation for IFN-g production (data not shown). Instead, our data indicate that IL-1 β promotes APC priming to promote CTL response (Supplementary Fig. 6m,n), similar to the reported mechanism in the infection model ¹¹.

- The authors state that all experiments were performed in accordance with their institutional ethics requirements. It is important to note that the tumors were allowed to grow to very large sizes (up to ~7000mm³). With most other studies required to cull mice at much smaller tumor sizes, it is difficult to put the results into perspective. In fact, when looking at tumors sizes smaller than 2000mm³, which is well beyond the end point of the majority of studies, the differences between the experimental conditions appear much less pronounced (Fig. 5c, g).

The point is well taken.

We just realized that there are different kinds of formulae to calculate tumor size. In our original submitted manuscript, we followed the few reference papers^{12, 13} to use the tumor size calculation formula (width \times length \times (width + length) \times 0.5) to show tumor size in our figures (see below, Fig. 5c-ori, Fig. 5g-ori, Fig. 5j-ori and Supplementary Fig. 5e-ori,f-ori). However, width \times width

$\times \text{length} \times 0.5$ is the most used tumor size calculation formula in the literature^{14, 15, 16, 17, 18, 19, 20, 21}. The tumor size data re-calculated by the formula ($\text{width} \times \text{width} \times \text{length} \times 0.5$) are shown in Fig. 5c,g,j and Supplementary Fig. 5e,f (see below). We found that the tumor size re-calculated by this commonly used formula is less than half of the size in our original submitted manuscript. Actually the maximum tumor size (about 3000 mm^3) in our data is smaller than those in some papers^{14, 16, 18, 19, 21} by the same calculation formula.

Figure 5. (c) Tumor size from wild-type, *Nlrp3*^{-/-}, *Casp1*^{-/-} or *beta2M*^{-/-} mice that were first injected s.c. with EG7 cells and then injected i.v. with OT1-CTLs.

Figure 5. (g) Tumor size in tumor from mice that were first injected s.c. with EG7 cells and then injected i.v. with OT1-CTLs or *Prf1*^{-/-} OT1-CTLs.

Figure 5. (j) Tumor size from wild-type and *Il1r1*^{-/-} mice that were first injected s.c. with EG7 cells and then injected i.v. with OT1-CTLs.

Supplementary Figure 5. (e, f) Tumor sizes in wild-type mice first injected s.c. with EG7 (EL4-ova) (e) or EL4 (f) tumor cell lines and then injected i.v. with OT1-CTLs for the indicated time.

As you can judge from the tumor weights which were not too big as shown in Fig. 5d,h,k, the tumor size relative to tumor weight was over-calculated by the formula (width × length × (width + length) × 0.5) in our original submitted manuscript. So we prefer to use the commonly used tumor size calculation formula (width × width × length × 0.5) to show our tumor size data. Actually there are dramatic and statistic differences within maximum tumor size of 2000 mm³ in Fig. 5c,g by the commonly used tumor size calculation formula.

So we now show all the data (Fig. 5c,g,j and Supplementary Fig. 5e,f) with the commonly used calculation formula. We changed the tumor size calculation formula in the M&M section accordingly.

- In Fig. 5b, the tumors should still express MHC-I molecules in the b2M^{-/-} mice. So unless, the authors injected naïve OT-I and thus required antigen presentation by the host (see my point on the kinetics above), it is not clear why antigen recognition on the tumor cells doesn't induce IL-

1b.

We are sorry for not describing this point clearly. We described that the T cell derived tumor cell line did not express inflammasome components (NLRP3 and pro-IL-1 β) (Supplementary Fig. 5l) and secrete IL-1 β (Fig. 5l and Fig. 5m,n). Therefore, even though the tumor cells can express MHC-I and present OVA antigen and response to OT-I CTLs, they can not secrete IL-1 β . All the host IL-1 β producing cells (mainly DC and macrophage cells) as shown in Fig. 5o,p lacked β 2m and thus did not respond to OT-I CTLs to process pro-IL-1 β for mature IL-1 β secretion.

- The first paragraph of the discussion needs to better reflect that the described phenomenon played no role in the infectious models tested here and therefore is not a generalizable principle.

The point is well taken. We have rewritten the first paragraph of the discussion as suggested. Although our in vitro data showed that alloantigen, tumor antigen or pathogen antigen specific CTLs all induced IL-1 β secretion in APCs (Fig. 1, Fig. 6a,b and Fig. 7a,b), the CTL-induced IL-1 β secretion is specifically important in the models of GVHD and antitumor but not in the infection models. As we discussed, we feel this specific phenomenon (not a generalizable phenotype) is interesting because in the situations of antigen specific antitumor immunity or GVHD where innate immunity is not so strongly activated as in infections, antigen specific CTL-induced IL-1 β secretion in APCs is critical for amplifying the antigen-specific effects.

Overall, I found this study well conducted and very interesting, although I am still somewhat skeptical about how well the observations in the in vivo tumor/GvHD models reflect CTL-induced IL-1beta production by APC upon antigen recognition.

We like to thank the reviewer for the positive comments on our study.

We have the following evidence to support the observations in the in vivo tumor/GvHD models reflect CTL-induced IL-1 β production by APC upon antigen recognition.

1) Antigen specific CTLs induced IL-1 β secretion in APCs in the antitumor and GVHD models in vivo (Fig. 5b,e,i, Supplementary Fig. 5g and Fig. 6c,f).

2) β 2M (antigen presentation) was required for antigen specific CTL induced IL-1 β secretion in

the antitumor and GVHD models in vivo (Fig. 5b, e and Fig. 6c).

- 3) NLRP3 and caspase-1 were required for antigen specific CTL induced IL-1 β secretion in the antitumor and GVHD models in vivo (Fig. 5e and Fig. 6c).
- 4) NLRP3, caspase-1 and IL-1 β were required for antigen specific CTL-mediated anti-tumor immunity and GVHD pathogenesis (Fig. 5c,j and Fig. 6d,e,g-j).
- 5) Tem CTLs are the major population in the two models and Tem CTLs had highest expression of perforin and highest ability to induce IL-1 β secretion in APCs (Supplementary Fig. 5i,j and Supplementary Fig. 6g,h).
- 6) We now have data to show that perforin in antigen specific CTLs was required for IL-1 β secretion and associated functions in both in vivo models of antitumor and GVHD. We measured IL-1 β secretion in the tumor model by transferring WT OT1 or *Prfl*^{-/-} OT1 CTLs to EG7 tumor bearing mice and found deficiency of perforin in OT1 CTLs resulted in significantly reduced IL-1 β level and consequently increased tumor size and weight (Fig. 5g-i). Similarly in the GVHD model, deficiency of perforin in the transferred allo-CTLs dramatically reduced IL-1 β level in the affected tissues (liver and intestine), and consequently the GVHD pathogenesis (body weight drop and survival) (Fig. 6f and Supplementary Fig. 6e,f). In contrast, perforin was not important for IL-1 β induction in the LM and LCMV infection models (Fig. 7e,f).

P.S.: Before our previous submission, we got the data of the effect of perforin deficiency in CTLs in the GVHD model (Fig. 6f and Supplementary Fig. 6e,f) and the no effect of perforin deficiency in the LM and LCMV infection models (Fig. 7e,f). We bred *Prfl*^{-/-} mice with OT1 mice to get *Prfl*^{-/-} OT1 mice. During the revision period, we utilized the *Prfl*^{-/-} OT1 CTLs to check the effects of perforin deficiency in CTLs in the antigen specific antitumor models (Fig. 5g-i). We have now included all the data to support the in vivo effects of perforin in CTLs.

References

1. Rentenaar RJ, Vosters JL, van Diepen FN, Remmerswaal EB, van Lier RA, ten Berge IJ. Differentiation of human alloreactive CD8(+) T cells in vitro. *Immunology* **105**, 278-285 (2002).
2. Voskoboinik I, *et al.* Calcium-dependent plasma membrane binding and cell lysis by perforin are mediated through its C2 domain: A critical role for aspartate residues 429, 435, 483, and 485 but not 491. *J Biol Chem* **280**, 8426-8434 (2005).
3. Yagi H, *et al.* Structural Basis for Ca²⁺-mediated Interaction of the Perforin C2 Domain with Lipid Membranes. *J Biol Chem* **290**, 25213-25226 (2015).

4. Thiery J, *et al.* Perforin pores in the endosomal membrane trigger the release of endocytosed granzyme B into the cytosol of target cells. *Nat Immunol* **12**, 770-777 (2011).
5. Lee GS, *et al.* The calcium-sensing receptor regulates the NLRP3 inflammasome through Ca(2+) and cAMP. *Nature*, (2012).
6. Schrader-Fischer G, Paganetti PA. Effect of alkalizing agents on the processing of the beta-amyloid precursor protein. *Brain Res* **716**, 91-100 (1996).
7. Christensen KA, Myers JT, Swanson JA. pH-dependent regulation of lysosomal calcium in macrophages. *J Cell Sci* **115**, 599-607 (2002).
8. Hornung V, Latz E. Critical functions of priming and lysosomal damage for NLRP3 activation. *Eur J Immunol* **40**, 620-623 (2010).
9. Jo EK, Kim JK, Shin DM, Sasakawa C. Molecular mechanisms regulating NLRP3 inflammasome activation. *Cell Mol Immunol* **13**, 148-159 (2016).
10. Hornung V, *et al.* Silica crystals and aluminum salts activate the NALP3 inflammasome through phagosomal destabilization. *Nat Immunol* **9**, 847-856 (2008).
11. Pang IK, Ichinohe T, Iwasaki A. IL-1R signaling in dendritic cells replaces pattern-recognition receptors in promoting CD8(+) T cell responses to influenza A virus. *Nat Immunol* **14**, 246-253 (2013).
12. Boissonnas A, *et al.* Antigen distribution drives programmed antitumor CD8 cell migration and determines its efficiency. *J Immunol* **173**, 222-229 (2004).
13. Boissonnas A, Fetler L, Zeelenberg IS, Hugues S, Amigorena S. In vivo imaging of cytotoxic T cell infiltration and elimination of a solid tumor. *J Exp Med* **204**, 345-356 (2007).
14. Rutkowski MR, *et al.* Microbially driven TLR5-dependent signaling governs distal malignant progression through tumor-promoting inflammation. *Cancer Cell* **27**, 27-40 (2015).
15. Johnston RJ, *et al.* The immunoreceptor TIGIT regulates antitumor and antiviral CD8(+) T cell effector function. *Cancer Cell* **26**, 923-937 (2014).
16. Yang X, *et al.* Targeting the tumor microenvironment with interferon-beta bridges innate and adaptive immune responses. *Cancer Cell* **25**, 37-48 (2014).
17. Rubinstein N, *et al.* Targeted inhibition of galectin-1 gene expression in tumor cells results in heightened T cell-mediated rejection; A potential mechanism of tumor-immune privilege. *Cancer Cell* **5**, 241-251 (2004).
18. Li X, Mao Q, Wang D, Xia H. A novel Ad5/11 chimeric oncolytic adenovirus for improved glioma therapy. *Int J Oncol* **41**, 2159-2165 (2012).
19. Panigrahy D, *et al.* Inhibition of tumor angiogenesis by oral etoposide. *Exp Ther Med* **1**, 739-746 (2010).
20. Favia A, *et al.* NAADP-Dependent Ca(2+) Signaling Controls Melanoma Progression, Metastatic Dissemination and Neovascularization. *Sci Rep* **6**, 18925 (2016).
21. Poeck H, *et al.* 5'-Triphosphate-siRNA: turning gene silencing and Rig-I activation against melanoma. *Nat Med* **14**, 1256-1263 (2008).

The following are the changes of figure re-organization in the revision compared to the previous/original submission:

1) Re-organization of the figures in the original submission

The control was added in Figure 2j (bottom panel); The unnecessary data with NH₄Cl treatment were removed in Figure 4g and Supplementary Figure 4d.

The bar plots were changed to dot plots in Figure 2j, Figure 3j,k, Figure 5a,b,n,p and Supplementary Figure 5b,c, according to the journal data format requirement.

The tumor size was re-calculated in Figure. 5c,g,j and Supplementary Figure. 5e,f.

Figure 4j,k was changed to Figure 4l,m; Figure 5g-m was changed to Figure 5j-p; Figure 6f-i was changed to Figure 6g-j; Figure 7e,f was changed to Figure 7g,h; Supplementary Figure 5h-k was changed to Supplementary Figure 5i-l; Supplementary Figure 6e-l was changed to Supplementary Figure 6g-n.

2) Newly added figures in the revision

Figure 1l; Figure 4j,k; Figure 5 g-i; Figure 6 f; Figure 7 e,f; Supplementary Figure 1f; Supplementary Figure 4h,i; Supplementary Figure 5h; Supplementary Figure 6e,f.

All the changes are highlighted in bold fonts in the revised manuscript text file and the supplementary text file.

We would like to thank the reviewers again for their positive comments on our study and for the important questions and comments to strengthen our manuscript. We have carefully revised the manuscript according to the referees' comments and your instructions, and hope that our revised manuscript is now suitable for publication in Nature Communications.

Thank you for your great effort.

Sincerely,

Youcun Qian, Ph.D.

March 22, 2017

Dear Editor,

Enclosed please find our revised manuscript entitled "Antigen-specific CD8+ T cell feedback activates NLRP3 inflammasome in antigen-presenting cells through perforin" (NCOMMS-16-22387A). We thank all the reviewers for the insightful and positive comments. We appreciate the opportunity you offered us to revise our manuscript. The followings are our point-by-point responses.

REVIEWERS' COMMENTS:

Reviewer #1 (Remarks to the Author):

The authors have responded well to my comments. I have no further questions.

We thank the reviewer for the positive comments.

Reviewer #3 (Remarks to the Author):

The authors have addressed my concerns sufficiently. I have to say though that the withdrawal of the NH₄CL data and the obviously grave mistakes in calculating tumor size are disappointing. I was more enthusiastic about the work before.

We thank the reviewer for the positive comments. As we explained before, we removed the NH₄CL data because they are not necessary and we could not exclude potential side effect of NH₄CL asked by the reviewer. We are sorry for not calculating tumor size with the most commonly used formula during our first submission.